# Improving the Methods for Processing Hard Rock Aquifers Boreholes' Databases. Application to the Hydrodynamic Characterization of Metamorphic Aquifers from Western Côte d'Ivoire

Kouassi Aristide Aoulou [1,2,*], Severin Pistre [2], Yéï Marie Solange Oga [1], Benoît Dewandel [3,4] and Patrick Lachassagne [2] 

[1] UFR-STRM, Université Félix Houphouët-Boigny, 22 BP 582, Abidjan 22, Côte d'Ivoire; ogamariso@gmail.com
[2] HSM, Univ. Montpellier, CNRS, IRD, Montpellier, France; severin.pistre@umontpellier.fr (S.P.); patrick.lachassagne@ird.fr (P.L.)
[3] BRGM, Univ. Montpellier, 34000 Montpellier, France; b.dewandel@brgm.fr
[4] G-eau, UMR 183, INRAE, CIRAD, IRD, AgroParisTech, Supagro, BRGM, 34000 Montpellier, France
* Correspondence: aoulouaristide08@gmail.com

**Abstract:** Statistical analysis of a borehole database, linear discharges, and water strikes processing enabled an understanding of the structure, geometry and hydrodynamic properties of the metamorphic hard rock aquifers from the Montagnes District, Western Côte d'Ivoire. The database comprises 1654 boreholes among which 445 only were usable for this research work after its pre-processing. Analysis shows that the structure of the aquifer is similar to that observed in several other areas in the world: it developed due to weathering processes, comprises the capacitive saprolite, 10–20 m thick on average, and an underlying transmissive fractured layer, overlying the unweathered impermeable hard rock. The fractured layer is 80 m thick, the first 40 to 45 metres being its most productive zone, with a 11.3 m$^3$/h median productivity. This research shows that metamorphic aquifers exhibit similar aquifer properties (thickness, hydrodynamic parameters) as plutonic ones and that there is interest in using such databases for research and other purposes. However, a rigorous pre-treatment of the data is mandatory, and geological data from published maps must be used instead of the geological data from the database. A previous methodology aiming at processing the boreholes' linear discharges was improved. It notably appears that the slope method must be preferred to the percentile method.

**Keywords:** groundwater; hydrogeological conceptual model; linear discharge; saprolite; mining impacts; water strike; weathered fractured layer

## 1. Introduction

Water resources (surface and groundwater) are essential for life, ecosystems and all economic activities. They are required for domestic, industrial, and agricultural needs (human food). Both for reasons of quantity, particularly during periods of low water levels when surface water may not be available, and for reasons of quality (lower treatment costs than surface water), groundwater is being used more and more frequently throughout the world [1]. It is now almost systematically utilised in African countries [2] and frequently in hard rock contexts. Hard rocks (HR) are plutonic and metamorphic rocks, from which we exclude marbles, as they can be karstified, and obviously limestones and non-metamorphosed volcanic rocks [1]. Hard rock aquifers generally occupy the first tens of metres below ground surface [3].

Almost half of the African continent's population relies on groundwater [4]. In many rural areas of sub-Saharan Africa, groundwater is the only sustainable source of water for human consumption. In Côte d'Ivoire, groundwater is the main source of drinking water for the rural population. Mining activities in the country are expanding rapidly

due to the large and underdeveloped mineral resources and government incentive policy. These activities can pose threats to groundwater quantity and quality. Therefore, a good understanding of the structure and functioning of aquifers in these mining regions is needed to prevent these effects and to develop mining activities that are as environmentally friendly as possible.

However, almost the entire territory of Côte d'Ivoire, 97.5% [5], is made up of crystalline rocks, and so these mining activities concern hard rock (HR) regions [1].

The conceptualisation of the aquifers of these HR regions in terms of structure (geometry, hydrodynamic properties) and functioning (recharge, flow, and groundwater discharge) is the subject of scientific debate, particularly with regard to the origin of their permeability and their fracturing [6,7].

Previously considered mainly as being tectonic in origin [8–12] with fractures mostly subvertical, it has now been demonstrated in most regions of the world that this permeability is due to the development of stratiform weathering profiles several tens of meters thick (see in particular [1,13], who propose a detailed bibliographical review of these different concepts and the presentation of associated hydrogeological conceptual models). This conceptual model is briefly described in Section 2. Similar results were also obtained on different types of rocks, such as carbonate rocks [14–16].

In the context of this scientific debate, this study aimed to determine whether this concept of weathering-related fracturing can also be applied in Côte d'Ivoire, where several authors [12,17–19] have recently used "tectonic" concepts. The consequences of using one or the other model are very significant. If we take the example of assessing the impact of a mining activity, in the case of the fracturing model linked to weathering, the impact will generally be limited to the topographic catchment area of the mine and the watercourses that flow from it. In the case of the tectonic fracturing model, impacts could be envisaged over long distances, for example, along "regional fracturing corridors". The potential consequences of these two conceptual models are detailed in the Section 4 of this paper.

Furthermore, concepts on the structure and functioning of basement aquifers have mainly been developed within granitic rocks, whose geological structure—generally homogeneous and isotropic, and therefore the hydrogeological structure—is simpler than that of metamorphic rocks (see for example [1,20]). The latter shows mineralogical variations, foliation, or schistosity, etc. As a result, they are more complex and have been much less studied [1,13]. This research therefore also aimed to improve our knowledge of the structure and functioning of aquifers in metamorphic rocks.

Finally, this study was mainly based on the use and processing, by means of various statistical methods, of a hydrogeological database of several hundred boreholes made available by the Office National de l'Eau Potable (ONEP) and concerning Western Côte d'Ivoire. Such databases have already been used in previous research projects and have shown their value (see for instance [3,21–23]), despite the relatively basic nature of the data available (total depth of the borehole, lithology, depth of the base of the loose alterite (saprolite), discharge at the end of the drilling, number and depth of the main water strikes, piezometric level, etc.).

Nevertheless, new methods for processing these data have recently been developed, based on the concept of stratiform weathering profile presented above [24]. The present study therefore also aimed to improve this methodology, in particular thanks to a more precise implementation and subsequent improvements, discussions, and clarifications of this method [24].

With these different objectives in mind, our research plan presents, to begin with, (i) the dataset used and the methodologies implemented; (ii) it also indicates the points on which methodological improvements were made; (iii) next, the results obtained are presented; (iv) finally, the results are discussed and compared with results already obtained elsewhere in the context of hard rocks. Consequences in terms of the hydrogeological impact of mining activities are drawn.

## 2. Materials and Methods

The data used in the present research are from a database of village hydraulic boreholes set up by ONEP for the Western Côte d'Ivoire. This section describes in turn: (i) the material available, i.e., the location and main characteristics of the study area, the type of data available in the ONEP database, and their main characteristics, and (ii) the statistical methods used to process them.

The Materials and Methods section also describes the available topographic data (digital elevation model, DEM) and the geological mapping used to support the interpretations of the borehole database.

The method developed by [24] will be used for the statistical processing of the available data.

The ONEP database was first critiqued to ensure the quality of the data available for this study. It was then statistically processed to produce a hydrogeological conceptual model of the Montagnes District aquifers.

Hydrogeologists are unanimous on the composite character of hard rock aquifers, which comprise (i) loose weathering or saprolite that is not permeable enough to provide significant discharge to common (small diameter) boreholes but which provides the capacitive function of the aquifer, and (ii) underlying fractures, tapped by most boreholes drilled in hard rock regions, which provide the transmissive function of the aquifer [1,9,20,25]. Figure 1 shows the conceptual model of such an aquifer [26], where the fractured layer is considered to be the lower part of the weathering profile. The unweathered underlying bedrock is impermeable. Very locally, it may have ancient tectonic fractures, joints, lithological contacts, veins, etc. whose permeability results from a local deepening of the weathering profile [1,27].

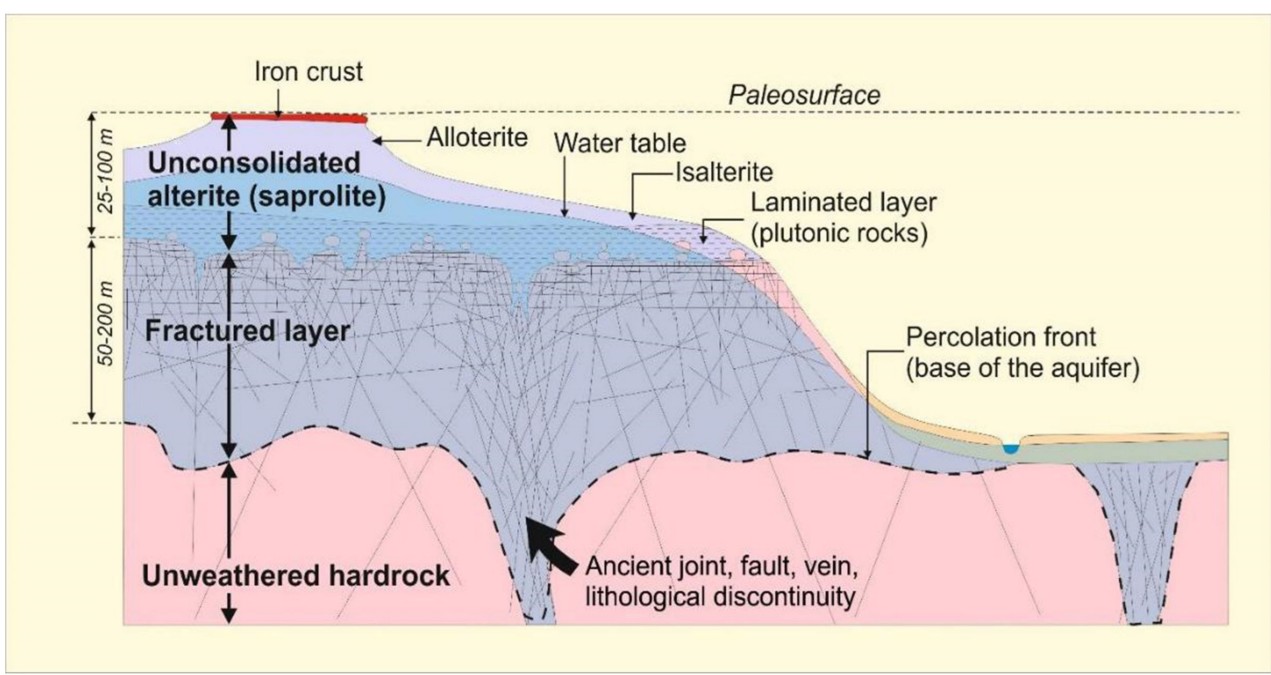

**Figure 1.** Conceptual model of a partly eroded paleo-weathering profile in hard rocks [26].

### 2.1. Material

2.1.1. Location and Main Characteristics of the Study Area

The Montagnes District lies in the Western Côte d'Ivoire (Figure 2). It is located between latitudes 5°50 and 8°00 N and longitudes 7°00 and 8°40 W. Covering an area of about 31,000 km$^2$, its capital is the city of Man. Its population is estimated at more than 2,400,000 inhabitants, with an average demographic density of 76 inhabitants/km$^2$ [28]. The main activities of the rural population are based on agriculture (rice, bananas, sweet

potatoes, cassava, cocoa, coffee, cotton, cola) and mining (industrial and artisanal), which has been developing significantly in recent years. The largest industrial gold mine in Côte d'Ivoire is at Ity in the commune of Zouan-Hounien, with an annual production of 6 to 7 tonnes. Fishing, livestock breeding, and tourism are secondary activities [29].

The hydrographic network is dense. Two major rivers border the study area, notably the Sassandra River to the East and the Nuon River to the West, the latter of which serves as the border between Liberia and Côte d'Ivoire. The climate, which is dependent on the Intertropical Convergence Zone (ICZ), is characterised by an annual rainfall of 1000 to 1500 mm. There are two seasons: a long rainy season lasting 8 months (March–October) and a dry season lasting 4 months (November–February).

The soils are generally ferralitic [30]. The natural vegetation is defined between the Guinean and Sub-Sudanese domains with mostly mesophilic and mountain forests.

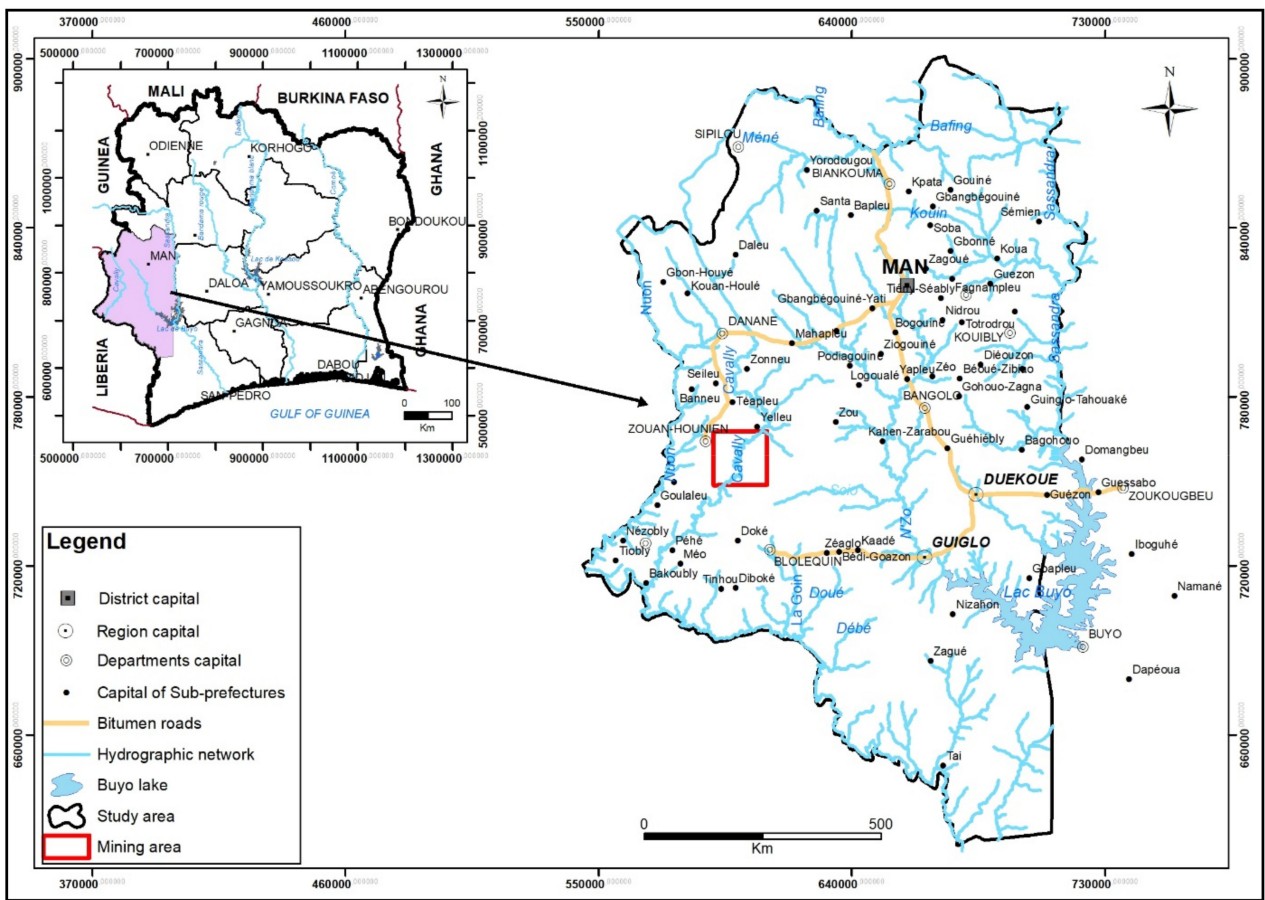

**Figure 2.** Location of the Montagnes District in the Western Côte d'Ivoire.

This region, commonly referred to as the Square Degree of Man, includes a mountainous part in the North, with altitudes sometimes exceeding 1000 m (Figure 3). The highest peaks are found at Momi (1302 m asl), Tonkpi (1189 m asl), Glanho (1175 m asl), and Mia (1077 m asl). The centre and the South constitute a vast plateau corresponding to a monotonous landscape with altitudes varying mostly between 250 and 350 m.

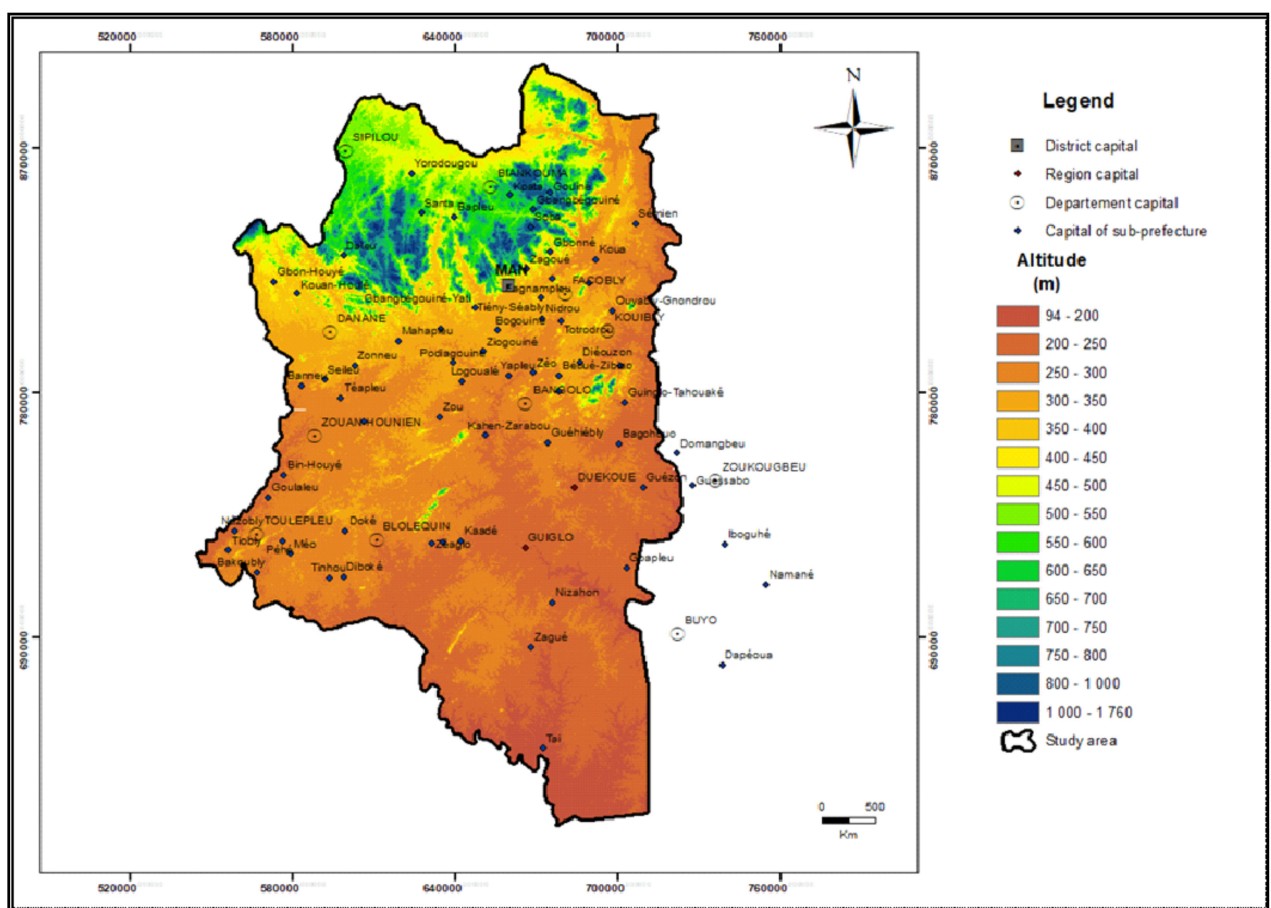

**Figure 3.** Topographic map of the Montagnes District in Western Côte d'Ivoire (Data source: http://earthdata.nasa.gov/ (accessed on 12 March 2020)).

### 2.1.2. Geology

The geological context of the Man region is part of the history of the West African craton in general and the Keneman-Man domain in particular. This region of Côte d'Ivoire was structured by two main orogenic cycles [31,32]: the Leonian megacycle (3500 Ma–2900 Ma) and the Liberian megacycle (2900 Ma–2600 Ma).

The geological formations are exclusively metamorphic and plutonic, of Archean age, with the exception of rare alluvial formations. The metamorphic rocks are composed of gneiss, amphibolo-pyroxenites, quartzites, micaschists, and migmatites. The plutonic rocks are of infracrustal origin and are represented by a complex of basic and ultrabasic rocks in the Man region, migmatites, charnockites, and granites associated with migmatites [33,34]. In addition, a charnockitic complex appears as intrusions in the granite-gneissic basement, or as anatectic mobilisates.

- Geological mapping

A geological map (Figure 4) representative of the formations encountered in the study area was produced from a compilation of two 1:200,000 geological maps provided by the Geology Directorate of the Ministry of Mines and Geology of Côte d'Ivoire [35,36]. These maps cover the northern and southern parts of the Montagnes District, respectively.

The two geological maps were not available in vector form, only in raster form. All geological contours were digitised (polygons), as well as were the faults (polylines).

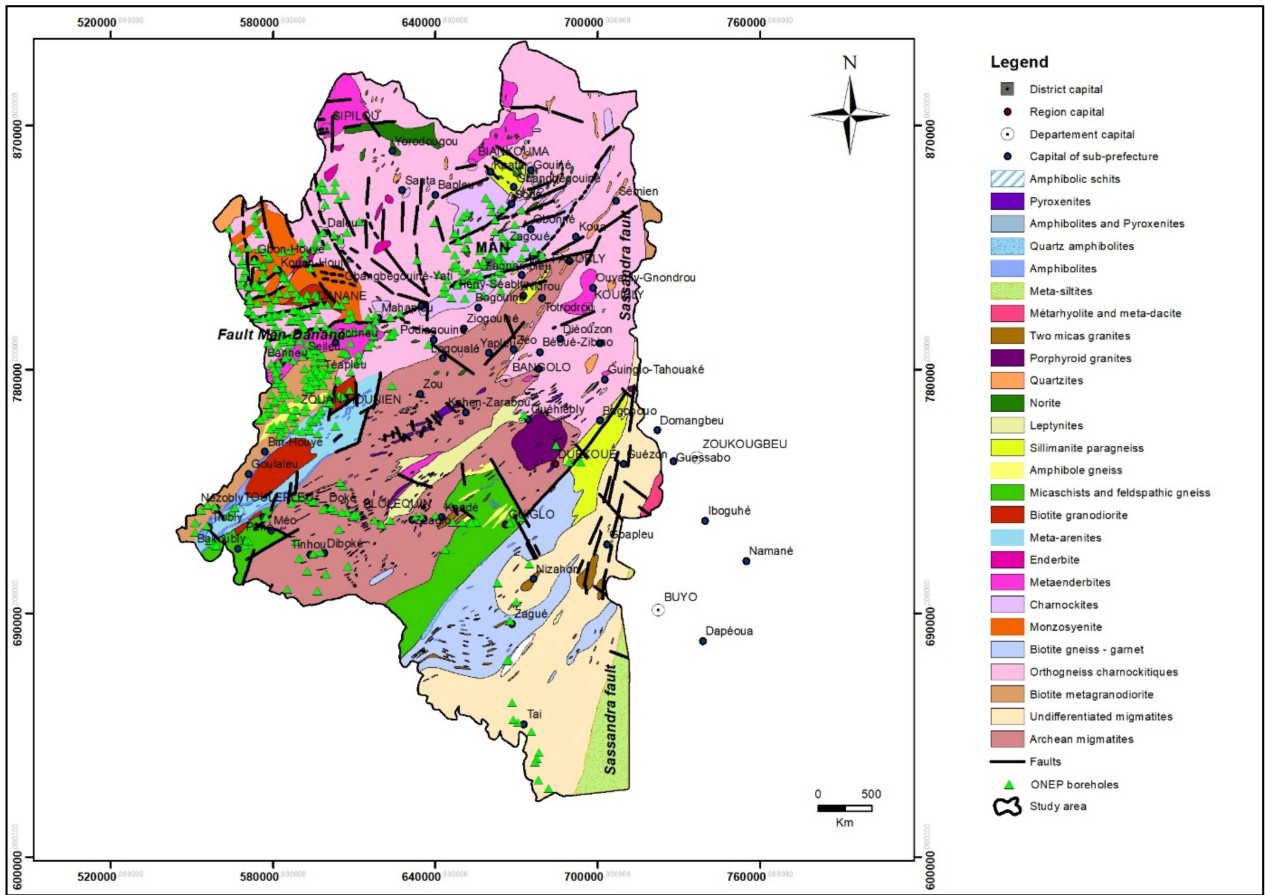

**Figure 4.** Geological map of the Montagnes District (west of Côte d'Ivoire), from [35,36]. The green triangles enable the location of the 445 boreholes which were used in the statistical analysis after eliminating those of poor quality.

- Grouping of rock units and associated boreholes

The resulting geological map (Figure 4) covering the Montagnes District comprises 26 rock units. Only 15 of these 26 rock units are intersected by the 445 boreholes in the ONEP database used in this work, as described in Section 3.2. In order to have statistically significant samples (sufficient numbers in each sample), groupings of rock units were made.

Therefore, the geological information available in the ONEP database and the geological map presented above were first compared and evaluated in terms of quality and reliability. The best and most reliable source of information was selected for the attribution of a lithology to each borehole. Second, based on the assumption that the hydrodynamic properties within a weathering profile are related to the mineralogy of the source rock and therefore that lithological units of similar texture and mineralogy have similar hydrogeological properties, the method of [24], described below in Section 2.2.4, was used as a grouping method to compare two groups of boreholes. First, the two groups were considered individually. Second, these two groups of boreholes were combined and considered as a unique set. Then, the validity of the grouping thus made was validated using the method of [24] if the grouped unique set was not statistically different from the two groups considered individually.

### 2.1.3. DEM

A Digital Elevation Model (DEM) of the Montagnes District was used in this study. It is from the Shuttle Radar Topography Mission (SRTM) altimetry data, created in February 2000 and downloaded from http://earthdata.nasa.gov/ (accessed on 12 March 2020). The SRTM data are provided in the form of grids, made up of squares of 1 arc degree (about 30 metres) sides, with geographic coordinates (X, Y, and Z) based on the WGS84 data.

From an image mosaic, 12 squares covering the study area were assembled in ARCGIS.10.2. (using the mosaic function) to provide a DEM coverage of the entire Montagnes District. The accuracy announced by NASA is $+/- 20$ m in planimetry (X and Y) and $+/- 16$ m in Z (for absolute heights). Checks were carried out, and they show that the DEM is relatively "smooth", with no significant unjustified variations between neighbouring meshes.

### 2.1.4. The ONEP Database

The database of boreholes from village hydraulic water supply programmes in hard rocks managed by the Office National de l'Eau Potable (ONEP) contains 1654 geo-referenced borehole records in the study area. Many countries in West and Sub-Saharan Africa have such databases (see for example: [3,22,24]). Each drilling data sheet contains the information (Table 1) collected, unless otherwise stated, by the companies (drilling companies, engineering offices, etc.) in charge of the drilling works or campaigns. The database used contains only quantitative data (no data concerning hydrochemistry or water quality). It does not include hydrodynamic parameters (transmissivity, permeability, etc.).

The boreholes were surely drilled during various different projects that are not explicitly cited in the database. Experience from previous studies shows that it does not impact the quality of the main data sets, such as depth of borehole, depth to the base of the saprolite, piezometric level, etc. [3,21–24]. We, however, show later on that this may have an influence on the geological description (see Section 2.1.2 of this paper).

The ONEP database was corrected for anomalies and sorted to remove duplicate boreholes with the same name and/or coordinates. The database resulting from this first process was then sorted to identify missing data. Boreholes without information on saprolite thickness and blowing discharge at the end of drilling were discarded from further processing.

**Table 1.** Description of the ONEP database.

| Parameter | Method | Unit | Reference | % of Boreholes with Information Available | Comments and Observations |
|---|---|---|---|---|---|
| Borehole location (X, Y) | GPS or by location on a map | m | WGS 84 Zone 29 North | 100 | Measurements are usually made in the field with a GPS or, for old drillings, by determination on a map. The accuracy on X and Y is estimated to be between a few meters and a few tens of meters (usual accuracy of a GPS). Location errors were identified in another work (Aoulou et al., in prep.). Nevertheless, since all statistical analyses in this work were performed relatively (depth relative to the ground surface), these errors were not detrimental to its quality. |
| Borehole elevation (Z) | GPS or from a topographic map | m | Above Sea Level | 60 | Measurements are usually made in the field with a GPS or, for old drillings, by determination on a map. The accuracy on Z was estimated to be between a few meters and a few tens of meters (usual accuracy of a GPS). Location and elevation errors were identified in another work (Aoulou et al., in prep.). Nevertheless, since all statistical analyses in this work were performed relatively (depth relative to the ground surface), these errors were not detrimental to its quality. |
| Date of drilling | (-) | (-) | (-) | 39.4 | The oldest wells date from 1964 and the most recent from 1991. |
| Lithology | Identification from cuttings during drilling | (-) | (-) | 96 | For many boreholes, this information ("biotite cratonic granites", "unconformable granites", "geosynclinal granites", "migmatites + gneisses (Liberian)", "greenstone", "Birrimian schists", and "sedimentary rocks"; translated into English), as identified in the ONEP database, does not correspond to that of existing geological maps (see Chapter 3.2). Therefore, information from the 1:2,000,000 scale geological maps [35,36] provided by the Geological Directorate of Côte d'Ivoire was used in the statistical processing of the data. |
| Aquifer | Identification during drilling | (-) | (-) | 53 | The keywords indicated were: weathering, fractured basement (translated into English). |
| Type of well | (-) | (-) | (-) | 100 | All were boreholes |
| Condition of the work | (-) | (-) | (-) | 60 | The majority of the wells were operated (51.5%). The rest of the boreholes in the ONEP database were marked "negative boreholes" (5.2%) and "abandoned" (3.4%) concerning this parameter. |
| Total depth of the borehole | Measurement at the end of drilling | m | Surface of the ground | 100 | |
| Thickness of the saprolite | From cuttings and drilling parameters during drilling | m | Surface of the ground | 81 | |
| Depth of the Water strike | Identification during drilling | m | Surface of the ground | 83 (Water strike 1), 60 (Water strike 2). | The ONEP database only provided information on the depths of one or maximum two water strikes in each borehole. The first water strike ("Water strike 1") was shallower than the second ("Water strike 2"). The flow rate of the water strike was never provided. Anomalies were observed. For some boreholes, the depth of "Water strike 2" was mentioned without any "Water strike 1". Thus, in these cases, it was deduced that this borehole recorded only one water strike. "Water strike 2" was then considered to be the "Water strike 1". |
| Depth to the piezometric level | Piezometric level without pumping usually measured at the end of drilling | m | Surface of the ground | 88.3 | |
| Instantaneous discharge | Air lift flow measured at end of the drilling by blowing air under pressure at the bottom of the well | m³/h | | 49 | |

*2.2. Methods*

2.2.1. Pre-Processing of the ONEP Database

Because of the thickness of the saprolite and the instantaneous discharge (i.e., air lift discharge measured at the end of the drilling by blowing air under pressure at the bottom of the well and providing a first estimate of the aquifer's transmissivity (e.g., see [21]), being two key parameters in this study, a pre-processing was performed first. Thus, all the boreholes that did not provide information on these parameters were eliminated from the ONEP borehole database.

In addition, the lithology reported in the ONEP data sheets associated with each borehole was found to be sometimes inconsistent with the existing geological maps [35,36] of the study area. Given this important information for this study, a second treatment allowed the appropriate geology from [35,36] to be assigned to each borehole in order to have geological data for each borehole in phase with the reality of the geological environment of the study area.

Thus, a database of boreholes made up of samples based on data on the thickness of the saprolite and the instantaneous discharge on the one hand and on lithology on the other (groupings by lithology according to the information; see Section 2.1.2) was obtained for the continuation of this work.

2.2.2. Elementary Statistics

Elementary statistics and graphic representations about the main parameters of the ONEP database were computed, such as histograms, averages, medians, and standard deviations, to enable the presentation and discussion of the elementary data.

2.2.3. Processing of Water Strikes Depth Data

The ONEP database contains information on the depth of water strikes observed during drilling. Water strikes are exclusively observed within the fractured layer (below the base of the saprolite) (Table 1). They are interpreted as being related to the existence of permeable fractures within the fractured layer of the hard rock aquifer. The ONEP database only contains, at best for each borehole, information on the depth of two water strikes, identified as water strike N°1 (WS1) and water strike N°2 (WS2) (see also Table 2).

**Table 2.** ONEP database of boreholes used in this study.

|  | Number of Boreholes | Percentage (%) | Relative Percentage (%) |
| --- | --- | --- | --- |
| Total number of boreholes in ONEP database | 1654 | 100 | NA |
| Duplicates | 624 | 37.7 | NA |
| Database without duplicates | 1030 | 62.3 | 100 |
| Boreholes without duplicates and without data about saprolite thickness | 316 | 19 | 30.7 |
| Boreholes without duplicates and without data about instantaneous discharge | 269 | 16.3 | 26 |
| Boreholes without duplicates with data about saprolite thickness and with instantaneous discharge | 445 | 27 | 43.3 |

To begin with, for each type of water strike (water strike N°1, water strike N°2), the depth of the water strike under the saprolite was calculated as the difference between the depth of the water strike considered and the depth of the base of the saprolite at the considered borehole. Then, for each lithological unit, the vertical distribution of the water strikes was studied using the graphical representations of the number of water strikes N°1 (WS1) and N°2 (WS2) under the base of the saprolite. These distributions were compared with the distribution of the depths of all the water strikes combined (1 and 2). This approach

made it possible not only to verify whether water strike 1 was exclusively observed above water strike 2 (in the ONEP database) but also to highlight the existence of a fractured layer under the saprolite of the hard rock aquifers in the study region and to characterise this fractured layer.

However, in order to reduce the likely influence of statistical bias that the lack of data in the ONEP database for deep boreholes might have on the interpretation of the frequency of water strikes under the saprolite, a quality ratio was calculated. The range of acceptability was chosen to be above 50%, according to the work of [20]. The number of boreholes through a given depth expressed as a percentage (quality ratio), in which the maximum number of water strikes below the saprolite was recorded, was determined according to Equations (1) and (2) below:

$$Xi\ (\%) = \frac{Number\ of\ borewells\ in\ a\ depth\ range}{\sum(Number\ total\ of\ borewells)} * 100 \tag{1}$$

where $Xi$ is the percentage (%) of boreholes that reach a given depth.

The quality ratio is:

$$Y(L) = \sum_{l=Lmin}^{l=L} (Xi) \tag{2}$$

where $Y(L)$ is the cumulative percentage of number of boreholes for all possible values of $L$ present in the data set of wells included in the considered lithology. This distribution was observed in relation to the frequency of occurrence of water strikes as a function of borehole depth under the saprolite.

### 2.2.4. Statistical analysis

The method was inspired by recent concepts of permeability related to weathering in hard rock aquifers and by the method proposed by [24]. One of the key outputs of the method was the characterisation (thickness, productivity) of the fractured layer of hard rock aquifers. Productivity is expressed in terms of linear discharge ($m^3/h/m$) and its variation was observed as a function of the depth of the borehole below the base of the saprolite.

- Linear discharge of each well;

On the basis of the conceptual model of hard rock aquifers (Figure 1), [24] defined a new parameter that they named "linear discharge" (Equation (3)):

$$q_i(l_i) = \frac{Q_i}{l_i} \tag{3}$$

where $Q_i$ is the instantaneous discharge ($m^3/h$) at well number $i$; $l_i$ is the length of the well $i$ below the base of the saprolite (m); $q_i(l_i)$ is the resulting linear discharge of the well $i$ ($m^3/h/m$), respectively.

- Quantitative characterization of the properties of the fractured layer;

The cumulative percentage of linear flow below the base of the saprolite in each lithology was determined and plotted as a cumulative curve from Equation (4) [24]:

$$p_q(L) = \sum_{l=Lmin}^{l=L} q_i(l) / \sum_{l=Lmin}^{l=Lmax} q_i(l) \tag{4}$$

where $q_i(l)$ is the linear discharge of well i calculated with the well length l below the base of the saprolite ($m^3/h/m$), and $p_q(L)$ is the cumulative percentage of linear discharge (%) calculated for the sample composed by wells whose length below the base of the saprolite is less than $L$(m). The cumulative percentage $p_q(L)$ is calculated for all possible values of $L$ present in the data set of wells included in the considered lithology, varying from

$Lmin$ to $Lmax$. By definition, the cumulative percentage varies from $p_q(Lmin) = 0\%$ to $p_q(Lmax) = 100\%$.

A graphical representation of the cumulative linear discharge as a function of the length l of the borehole under the base of the saprolite was carried out according to the model in Figure 5, as proposed by [24]. The curve obtained was associated, for reference, with the curve of the cumulative number of boreholes, classified and normalised in the same order and method. Within the curve of cumulative linear discharges thus obtained, [24] generally distinguishes a median part of the curve with a linear trend, with a relatively low slope, corresponding to the highest linear discharge "q", which allows the thickness and productivity of the most productive part of the fractured layer to be characterised. This part of the curve corresponds to the part of the fractured layer with the most densely permeable fractures (therefore the most transmissive part). Then, the linear discharges decrease when the base of the fractured layer is reached or even exceeded. This final part of the curve characterises the deepest part of the fractured layer and any underlying permeable discontinuities. The first part of the curve sometimes has a steeper slope at its beginning (Figure 5); the origin of which is not explained by [24]. We will attempt to explain its origin in this paper.

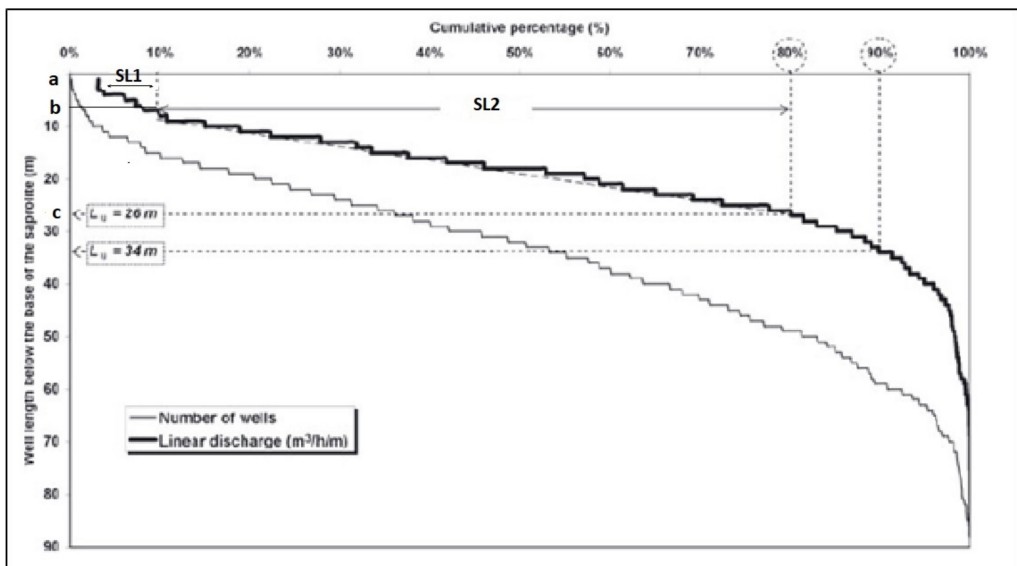

**Figure 5.** Cumulative percentage of linear discharge (X, %) as a function of the well length below the base of the saprolite (Y, m) to estimate the properties of the fractured layer for porphyritic biotite granite with 904 wells in Burkina Faso ([24] modified, Wiley-Blackwell Publishing, Inc., 10 November 2021).

For the purpose of this paper, the curve indicating the cumulative number of boreholes has not been shown. Only the cumulative linear discharge curve was considered for the characterisation of the parameters of the fractured layer. The evolution of the cumulative percentages of linear discharge under the saprolite for a given lithological unit has five parameters, named $a$, $b$, $c$, $SL1$, and $SL2$ (Figure 5):

Parameters "$a$" and "$b$" correspond to the beginning and end of the steep slope of the curve, if any ($SL1$; expressed in m/%), expressed as the depth of drilling below the base of the saprolite (m). The cause of this steeper slope is not explained by [24]; it will be studied in this paper;

Parameters "$b$" and "$c$" correspond to the beginning and end, respectively, of the linear part of the $SL2$ slope curve (m/%), expressed as the depth of drilling below the base of the saprolite (m);

Parameter "$c$" corresponds to the depth of the borehole below the base of the saprolite beyond which a decrease in linear discharge with depth is observed;

The parameter "*SL2*" is defined as the slope of the linear part of the curve, also expressed in m (m/%);

The parameter "*SL1*" corresponds to a fairly steep slope obtained for shallow drilling below the base of the saprolite; also expressed in m (m/%).

Second, for each lithology, the mean productivity of the most transmissive part of the aquifer (of the fractured layer) can be quantified through the Equations (5) and (6) [24]:

$$q_M(Lu) = \sum_{l=Min(li,\ i=1,\ n)}^{Lu} q_i(l) / j(Lu),\ i = 1,\ n \tag{5}$$

$$Q_M(Lu) = q_M(Lu) * Lu \tag{6}$$

where $L_u$ (m) is the "useful thickness" of the fractured layer (defined either through the slope or the percentile method, see above in [24]); $q_M(Lu)$ is the mean linear discharge (m$^3$/h/m) evaluated for the "useful thickness" $Lu$; $j(Lu)$ is the corresponding number of wells; and $Q_M(Lu)$ is the resulting mean discharge (m$^3$/h) for "theoretical" wells that would intersect the whole "useful thickness" (length $L_u$) of the fractured layer.

A study of the correlation between the *SL2* slope and $Q_M(Lu)$ was carried out for each of the lithological units.

## 3. Results

### 3.1. Pre-Processing of the ONEP Database

#### 3.1.1. Usable Boreholes

Table 2 shows that 37.7% of the boreholes in the ONEP database were duplicates. In addition, 19% and 16.3% of the boreholes did not have information on the parameters thickness of the saprolite and instantaneous discharge at the end of drilling, respectively. Thus, of the total of 1654 boreholes in the ONEP database, 445, or 27%, were usable and were used in this work. This dataset presented information on (1) total depth, (2) thickness of the saprolite; (3) the instantaneous discharge, (4) depth of the water strike N°1 and/or N°2, (5) depth of the piezometric level after drilling, and (6) geology.

The 1209 wells that were duplicates or without data about the saprolite thickness and instantaneous discharge were not considered in this study.

#### 3.1.2. Identification of the Lithological Units Associated with Each Borehole

The geological data recorded (Table 3) in the ONEP database showed many inconsistencies with the available geological maps. This information about geology was not available for 64 boreholes (3.8%). Sedimentary lithology was attributed to 214 boreholes (13%), although this type of geological formation is not recognised in this region, which, with the exception of rare alluvial formations, is composed exclusively of hard rocks. These 214 boreholes did not correspond to such alluvial formations; according to the geological map, they are located, by order of decreasing number, in feldspathic gneiss (56 boreholes), Archean migmatites (30 boreholes), undifferentiated migmatites (28 boreholes), etc.

**Table 3.** Lithologies of boreholes according to the ONEP database (translated from French).

| Lithological Description | Number of Boreholes | Percentage (%) |
|---|---|---|
| Biotite granite | 1001 | 60.5 |
| Migmatites and gneiss (Liberian) | 309 | 18.7 |
| Sedimentary | 214 | 13 |
| Birrimian Schist | 27 | 1.6 |
| Green Rocks | 22 | 1.3 |
| Discordant granites | 13 | 0.8 |
| Geosynclinal granites | 4 | 0.2 |
| Lithology not known | 64 | 3.9 |
| **TOTAL** | **1654** | **100** |

These anomalies and the almost consistent mismatch between ONEP database lithology and geological mapping show the unreliability of this ONEP information collected in the field, often by non-geologists.

Thus, in order to work with reliable lithological data, the lithology of each of the 445 boreholes studied was assigned on the basis of the geological mapping (Figure 4). Following these checks and corrections, 26 different lithologies were considered from the 445 boreholes selected (Table 4).

**Table 4.** Lithological units from the geological map and number of associated boreholes.

| Number | Lithological Description | Area (km$^2$) | % of the Study Area | Number of Boreholes | % of the Number of Boreholes |
|---|---|---|---|---|---|
| 1 | Charnokites | 1133.6 | 3.6 | 52 | 11.7 |
| 2 | Charnockitic orthogneiss | 10,803.2 | 34.8 | 93 | 20.8 |
| 3 | Biotite granodiorite | 475.5 | 1.5 | 14 | 3.2 |
| 4 | Biotite metagranodiorite | 1280.9 | 4.1 | 105 | 23.6 |
| 5 | Enderbites | 106.5 | 0.3 | 6 | 1.3 |
| 6 | Metaenderbites | 1103.6 | 3.5 | 41 | 9.2 |
| 7 | Undifferentiated migmatite | 3265.2 | 10.5 | 14 | 3.2 |
| 8 | Archean migmatites | 4307.7 | 13.9 | 38 | 8.5 |
| 9 | Monzosyenites | 710.2 | 2.3 | 34 | 7.6 |
| 10 | Meta-arenites | 1095.7 | 3.5 | 18 | 4 |
| 11 | Micaschists and Feldspathic gneiss | 1275.7 | 4.1 | 11 | 2.5 |
| 12 | Amphibole gneiss | 204.3 | 0.6 | 15 | 3.5 |
| 13 | Biotite gneiss-garnet | 1882.7 | 6.1 | 2 | 0.5 |
| 14 | Leptynites | 411.2 | 1.3 | 0 | 0 |
| 15 | Porphyroid granites | 288.6 | 0.9 | 1 | 0.2 |
| 16 | Two micas granites | 182.8 | 0.6 | 0 | 0 |
| 17 | Amphibolites | 120.6 | 0.4 | 0 | 0 |
| 18 | Quartz amphibolites | 45.5 | 0.2 | 0 | 0 |
| 19 | Amphibolites and Pyroxenites | 28.7 | 0.1 | 0 | 0 |
| 20 | Pyroxenites | 169.7 | 0.5 | 0 | 0 |
| 21 | Norite | 411.8 | 1.3 | 0 | 0 |
| 22 | Quartzites | 264.4 | 0.8 | 0 | 0 |
| 23 | Sillimanite Paragneiss | 745.7 | 2.4 | 1 | 0.2 |
| 24 | Amphibolic schists | 6.5 | 0.02 | 0 | 0 |
| 25 | Meta-siltites | 758.5 | 2.4 | 0 | 0 |
| 26 | Metarhyolite and Meta-dacite | 58.5 | 0.2 | 0 | 0 |
| | **TOTAL** | **30,972.58** | **100** | **445** | **100** |

Table 4 shows that 11 lithological units were not characterized by any boreholes from the ONEP database. Among the 15 other lithological units, some showed very low numbers, on which it was difficult to carry out representative statistical analyses. We therefore decided to group some lithologies together according to the principle described in Section 2.2.4, considering that lithologies can be assumed to be mineralogically very similar.

Figure 6 shows the cumulative percentage curve of the linear discharge as an example for the following lithological units:

a.  Charnockitic orthogneiss (N = 93) which, according to the geological map, correspond to metamorphosed Charnockites;

b.  Charnockites (N = 52);

c.  Charnockitic orthogneiss and Charnockites (N = 145).

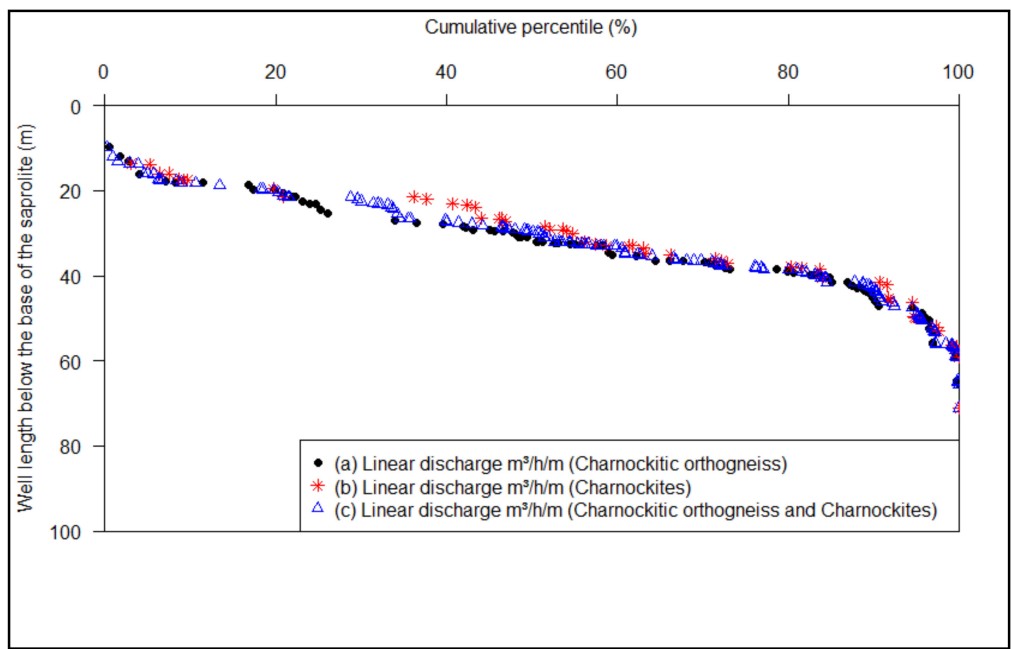

**Figure 6.** Cumulative percentage of linear discharge (X axis, %; see definition of the linear discharge in Section 2.2.4) as function of the length of the well below the base of saprolite (Y axis, m) for the lithologies: (a) Charnockitic orthogneiss (N = 93), (b) Charnockites (N = 52) and (c) Charnockitic Orthogneiss and Charnockites (N = 145).

One can note that the three statistical distributions are similar. This comparison of distributions was carried out on the analogous lithologies with low numbers. The results are in all respects similar to Figure 6 for all groupings made (see Supplementary Materials). This process allowed the justification for grouping the boreholes into six lithological groups, resulting in a minimum number of 35 boreholes for the lithology Monzosyenites and Porphyroid granites (Table 5) and a maximum number of 145 boreholes for the lithology Charnockitic orthogneiss and Charnockites.

**Table 5.** Grouping of lithological units and number of associated boreholes.

| Original Lithology | Number of Boreholes | Combined Lithology | % of the Study Area | Total Number of Boreholes | % of the Number of Boreholes |
|---|---|---|---|---|---|
| Charnokites | 52 | Charnockitic orthogneiss and Charnockites | 38.4 | 145 | 32.6 |
| Charnockitic orthogneiss | 93 | | | | |
| Biotite granodiorite | 14 | Biotite granodiorite and Biotite metagranodiorite | 5.6 | 119 | 26.7 |
| Biotite metagranodiorite | 105 | | | | |
| Undifferentiated migmatite | 14 | Archean and Undifferentiated migmatites | 24.4 | 52 | 11.7 |
| Archean migmatites | 38 | | | | |
| Porphyroid granites | 1 | Monzosyenites and Porphyroid granites | 3.2 | 35 | 7.8 |
| Monzosyenites | 34 | | | | |
| Enderbites | 6 | Enderbites and Metaenderbites | 3.8 | 47 | 10.6 |
| Metaenderbites | 41 | | | | |
| Meta-arenites | 18 | Micaschists, all Gneiss and Meta-arenites | 16.7 | 47 | 10.6 |
| Micaschists and Feldspathic gneiss | 11 | | | | |
| Amphibole gneiss | 15 | | | | |
| Sillimanite paragneiss | 1 | | | | |
| Biotite gneiss-garnet | 2 | | | | |
| **TOTAL** | **445** | | **92.1** | **445** | **100** |

### 3.2. Elementary Statistics

Based on the combined lithology (Table 5), the statistics of the main parameters are presented in Figure 7 and Table 6.

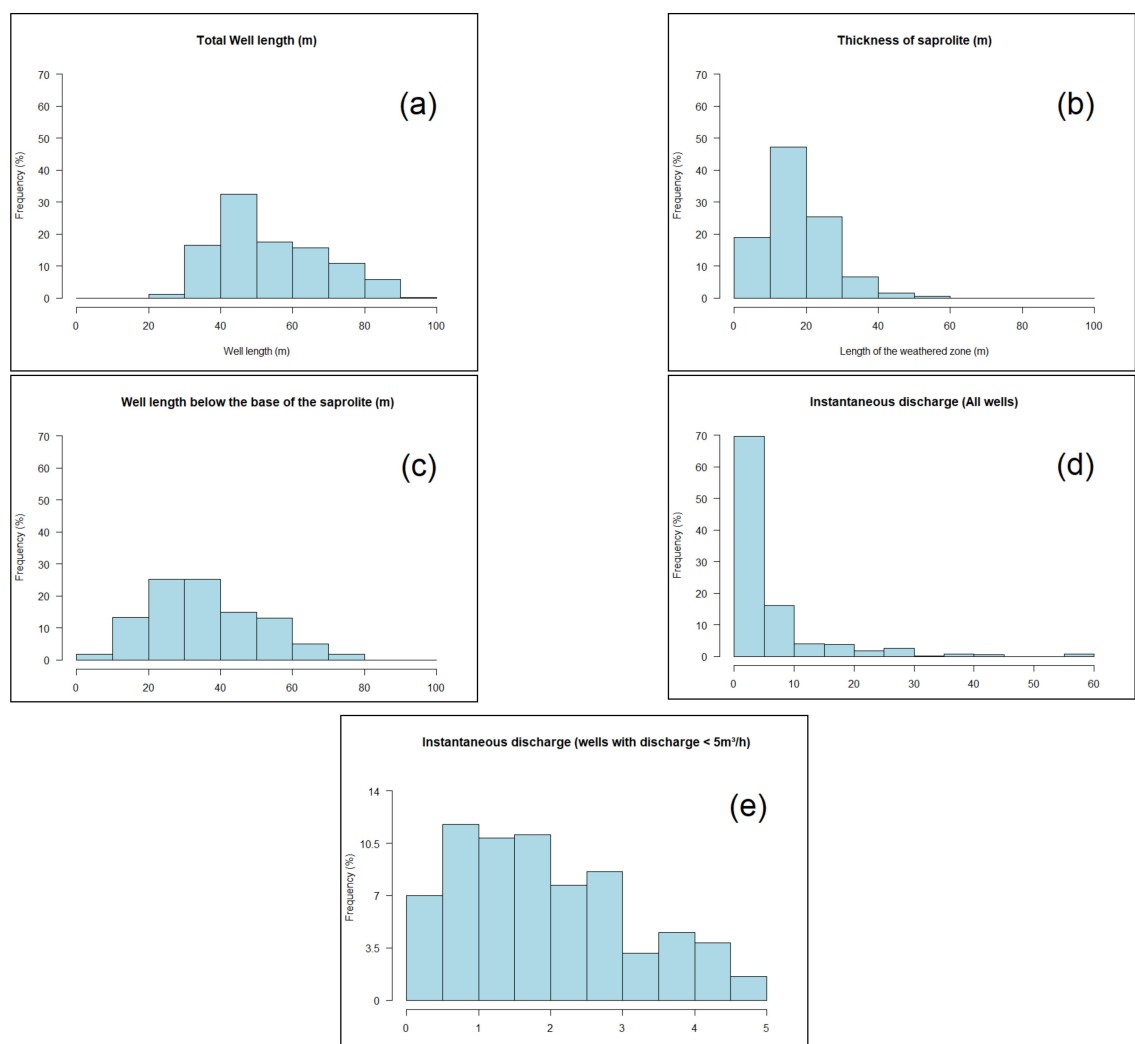

**Figure 7.** Histograms of the parameters from the database or computed from it (N = 445). (**a**) Total well length (X axis = well length (m), Y axis: frequency (%)); (**b**) thickness of the saprolite (X axis = length of the saprolite (m), Y axis: frequency (%); (**c**) well length below the base of the saprolite (X axis = well length under the base of the saprolite (m), Y axis: frequency (%)); (**d**) instantaneous discharge (all wells) (X axis = instantaneous discharge ($m^3$/h), Y axis: frequency (%)), and (**e**) instantaneous discharge (wells with discharge < 5 $m^3$/h) (X axis = instantaneous discharge ($m^3$/h), Y axis: frequency (%)).

**Table 6.** Elementary statistics computed from the ONEP database (N = 445).

| Parameters | Mean | Median | Standard Deviation | Most Represented Class |
|---|---|---|---|---|
| Total depth (m) | 53.7 | 50.2 | 14.6 | [40–50] |
| Thickness of the saprolite (m) | 17.9 | 16.9 | 8.7 | [10–20] |
| Well length below of the base of the saprolite (m) | 35.7 | 33.7 | 14.9 | [20–40] |
| Instantaneous discharge (<5 $m^3$/h) | 2 | 1.8 | 1.2 | [0.5–2] |
| Instantaneous discharge (<60 $m^3$/h) | 5.8 | 2.7 | 8.5 | [0–5] |

The total depth of boreholes shows a centred normal distribution with a similar mean and median (53.7 m and 50.2 m, respectively). The [40, 50 m] depth class is predominantly represented (32.3%).

The thickness of the saprolite ranges from 0.5 to 59.7 m. The average thickness of the saprolite is 17.9 m and the median is 16.9 m. The thickness class [10, 20 m[ is predominantly represented (47.8%) and few boreholes have less than 5 m of saprolite (2.7%).

The well length below the base of the saprolite is between 6 and 73.5 m. The absence of boreholes that tap only the saprolite is characteristic of hard rock aquifers. The saprolite is not permeable enough for small diameter boreholes such as a conventional diameter borehole to provide a significant discharge. The depth classes [20, 30 m[ and [30, 40 m[ are the most represented, with 24% and 26.3%, respectively.

The instantaneous discharge varies between 0 and 60 m$^3$/h and shows, and as is usual (see for example [6,18,24,37]) (Figure 7), what looks like a lognormal distribution. The mean discharge is 5.8 m$^3$/h. The median and standard deviation are 2.7 m$^3$/h and 8.5 m$^3$/h, respectively. The discharge rate class [0, 5 m$^3$/h[ is most represented (69.6%). Within this class, the discharge classes between 0.5 and 2 m$^3$/h are the most represented. This range of discharges is typical of fractured hard rock aquifers in the world, both in plutonic and metamorphic rocks: in Africa [3,18,22,24], in Korea [1,38,39], in India [20,25], and in Europe [1,37,40]. References on metamorphic rocks are nevertheless less abundant in the literature. Note that these discharge data are instantaneous discharge at the end of drilling and not the sustainable yield of the borehole, which is commonly much lower.

### 3.3. Characterisation of the Data Set

This chapter illustrates the results obtained for the two main lithologies present in the study area:

Charnockitic orthogneiss and Charnockites;
Biotite granodiorite and Biotite metagranodiorite.
The results for all other lithologies are summarised in the Supplementary Materials.

### 3.3.1. Linear Discharge of Each Well

Figure 8 shows the linear discharge expressed in reference to the depth of the borehole below the base of the saprolite for the two major lithological units in the study area. The linear discharge values are below 2 m$^3$/h/m, which is typical of hard rock aquifers [24,37,38,41]. The Biotite granodiorite/Biotite metagranodiorite group shows several extreme values (5 values > 1 m$^3$/h/m) in contrast to only 2 values > 1 m$^3$/h/m within the orthogneiss. We also note, as is usual [24,37], a tendency for the linear discharge to decrease exponentially with depth. For both lithologies, the highest linear discharges are mostly observed in the intervals [10, 50 m[ which could indicate that this is the most productive part of the fractured layer aquifer.

Such a graph shows the absence of data below 10–15 m, which agrees with the elementary statistics (Figure 7b,c) and which shows that all boreholes were drilled with a minimum depth of 10 m in the fractured layer below the base of the saprolite. Note that discharge rates were measured after total completion of the well.

### 3.3.2. Vertical Distribution of Water Strikes within the Wells

Water strikes were observed at depths between 0 and 70 m, with reference to the depth of the borehole below the base of the saprolite (Figure 9); no water strikes were observed within the last drilled interval, between 70 and 80 m (deepest water strike = 67.8 m below the base of the saprolite). Within the ONEP borehole database, only a maximum of two water strikes was recorded for each borehole. In addition, it was verified for all the boreholes with both a water strike (WS) 1 and a water strike 2 that the WS2 was systematically deeper than the WS1.

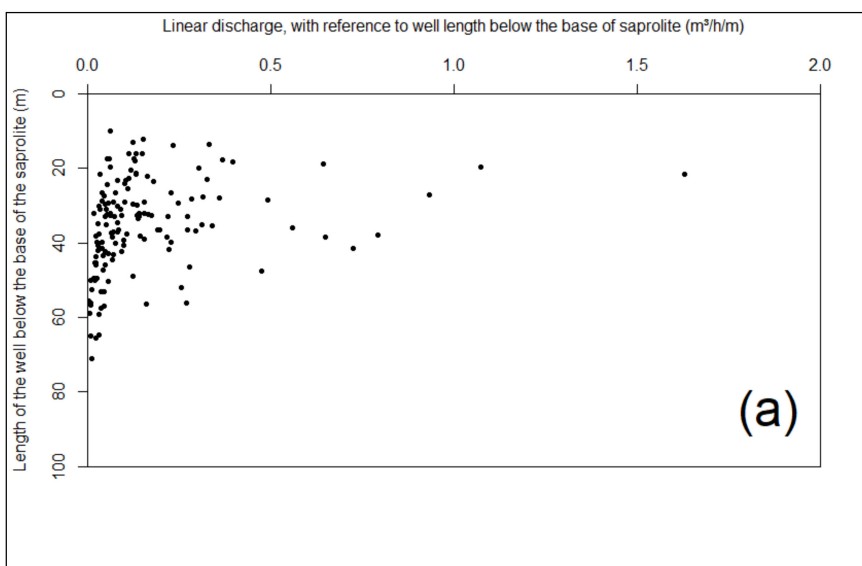

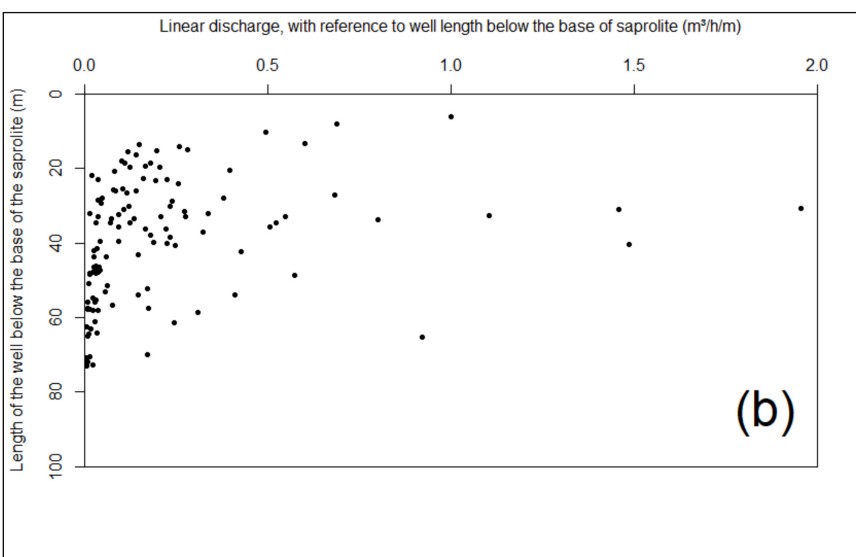

**Figure 8.** Linear discharges (X; m$^3$/h/m), computed with reference to well length below the base of the saprolite (Y; m), for the lithologies (**a**) Charnockitic orthogneiss and Chanockites (N = 145) and (**b**) Biotite granodiorites and Biotite metagranodiorite (N = 119).

Considering all lithologies combined, out of a total of 623 water strikes, the first water strike, or water strike N°1 (WS1), was observed at depths of between 0.3 and 62 m below the base of the saprolite. The second water strike (WS2) was observed at depths of between 1.7 and 67.8 m below the base of the saprolite. For all the boreholes, the [0, 10 m[ class (Figure 9a) contained most of the water strike N°1 data (35.5%) while the [10, 20 m[ class was the most numerous for water strike N°2 (23.8%). Below these depth classes, the numbers of WS1 and WS2 decreased in parallel. The total number of water strikes was recorded mostly in the first 40 m according to the following distribution: [0, 10 m[ (26.2%); [10, 20 m[ (25.5%); [20, 30 m[ (20.7%); [30, 40 m[ (15%). This represents a total water strike occurrence of 87.5% in the first 40 m below the saprolite. The calculated quality ratio of the number of boreholes reaching a given depth expressed as a percentage (Figure 9) for this portion of the hard rock aquifer in the region [0, 40 m[ was greater than 50%. This means that the most representative data were obtained in the first forty meters below the base of the saprolite. Beyond this depth into the aquifer, there was a decrease in the number of water strikes to the point of reaching a complete absence at depths greater than 70 m. The low observation rate (<50%) at larger depths (between 40 and 80 m) for WS1 and WS2

considered separately did not allow us to conclude with certainty that the decrease in the number of fractures could either be an indicator of the approach of the base of the fractured layer or be due to a lack of drilling data at such deep depth. Nevertheless, the significant data (≤40 m) for WS1&2 already showed a decrease in the number of water strikes within the first 40 m. This showed that the decreasing trend was significant. Therefore, it would be reasonable to assume that the decrease in the number of water strikes to a complete absence at depths greater than 70 m indicates that one is not far from the base of the fractured layer. In India and in granites, [20] have also observed an apparent decrease in the number of water strikes with depths. There the fractured layer was thinner; the decrease began from 15 m below the base of the saprolite.

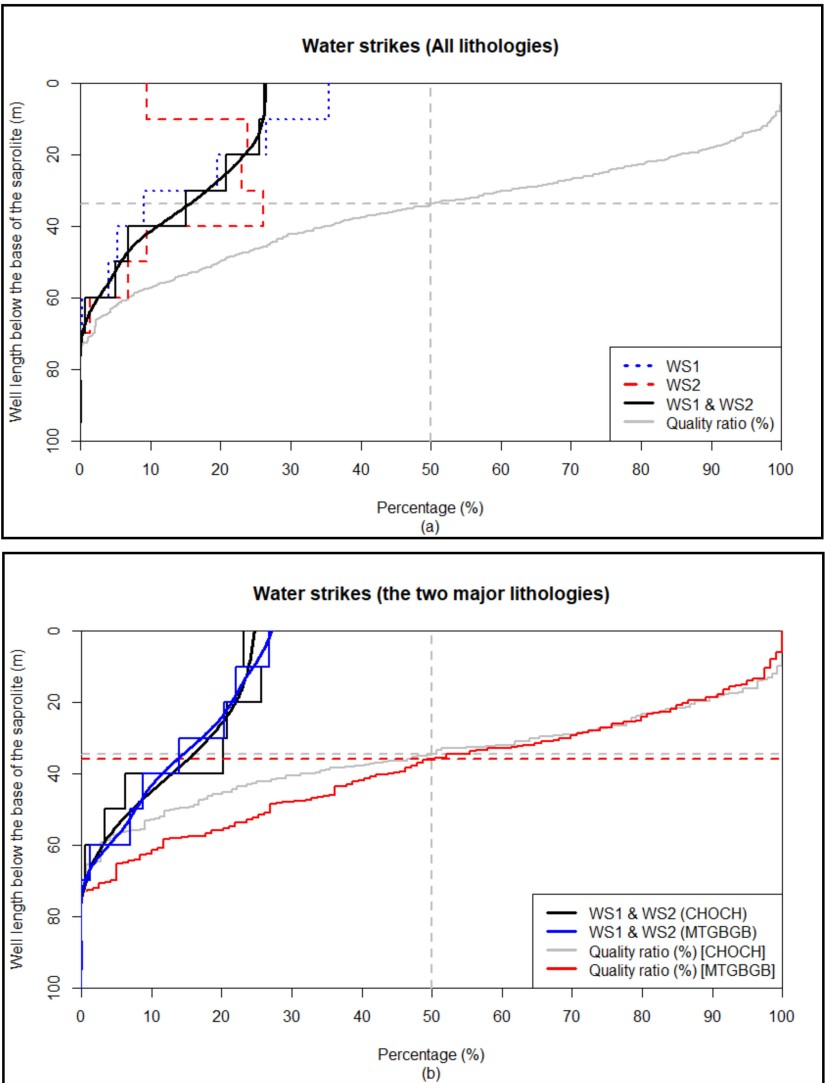

**Figure 9.** Vertical distribution of water strikes as a function of depth below the base of the saprolite in (**a**) all lithologies (NWS1 + NWS2 = 623; with NWS1 = 400 and NWS2 = 223), (**b**) Charnockitic Orthogneiss and Charnockites (CHOCH) (NWS1 + NWS2 = 190; with NWS1 = 129 and NWS2 = 61), and Biotite Granodiorite and Biotite Metagranodiorite (MTGBGB) (NWS1 + NWS2 = 171; with NWS1 = 107 and NWS2 = 64); WS1 and WS2 are respectively the first and second water inflow under the saprolite; Quality ratio is the ratio expressed as a percentage of the number of boreholes passing through a given depth.

Figure 9b shows the distributions of water strikes below the base of the saprolite for the Charnockitic orthogneiss and Charnockites, and for the Biotite granodiorite and Biotite metagranodiorite lithologies. The distributions are similar to that observed for all

the rock units present in the study area. The distribution profiles of the number of water strikes N°1 and N°2 recorded under the saprolite in the two major units as well as in all the lithological units studied (see Figures S3, S7, S11 and S14 in Supplementary Materials) in the framework of this work were similar.

These results are consistent with previous studies, which show that the first few meters or ten meters of the fractured layer are generally characterized by denser fracturing [20,26,38,42].

In our case, the fractured zone is about seventy meters thick, with the most fractured zone in the first 20 to 30 m, and then a decreasing trend.

The black and blue curves superimposed on the different histograms (Figures 8 and 9) are respectively the smoothed curves for WS1 + WS2.

### 3.3.3. Quantitative Characterization of the Properties of the Fractured Layer

Figure 10 shows the decrease in the linear discharge with depth below the base of the saprolite for the two lithologies chosen for illustration.

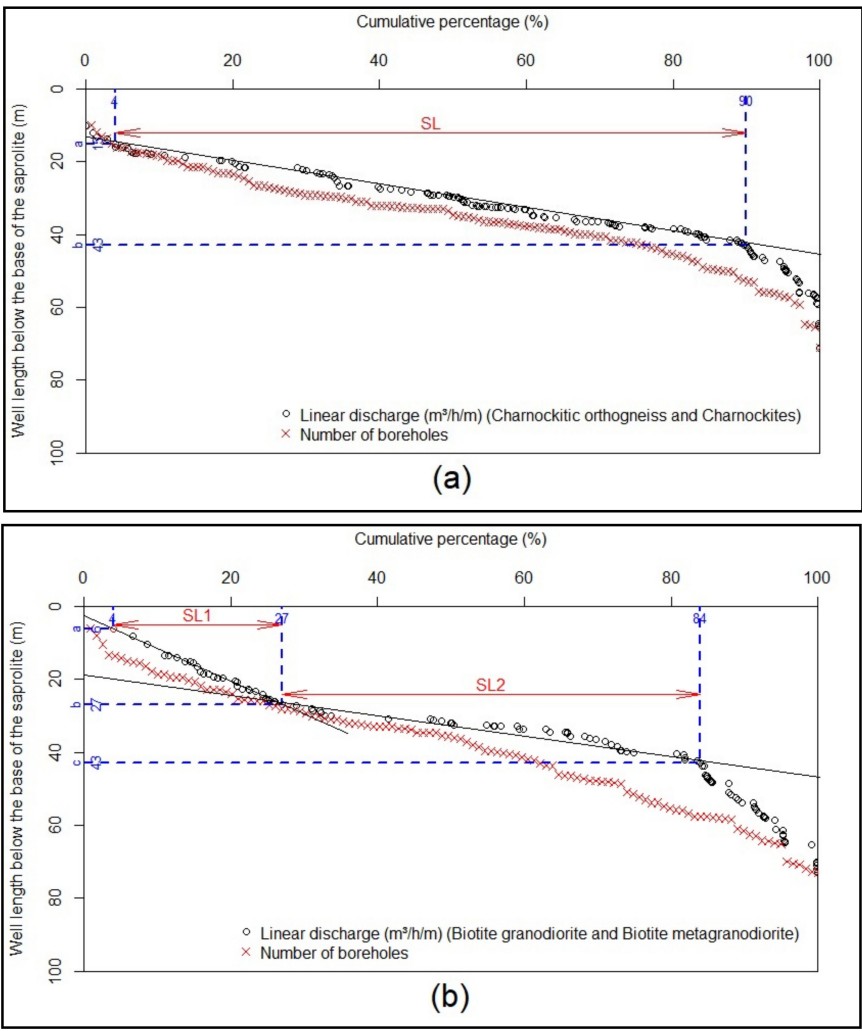

**Figure 10.** Cumulative percentage of linear discharge of the fractured layer for the lithologies (**a**) Charnockitic orthogneiss and Charnockites (N = 145) and (**b**) Biotite granodiorite and Biotite metagranodiorite (N = 119).

The absence of data for the shallowest depths of the boreholes below the base of the saprolite (the first 10 m) is logical and consistent with previous statistical treatments of the depth of the boreholes below the base of the saprolite and the calculation of linear discharge (Section 3.3). As the fractured layer was the target of the boreholes, boreholes in this area intersected it over at least 10 m.

The graph for the Charnockitic orthogneiss and Charnockites rock unit (Figure 10a) shows:

A first part, characterised by the absence of boreholes between 0 and 10 m, and 2 or 3 boreholes that did not allow a representative slope to be described.

A second part in the interval between 15 and 43 m below the base of the saprolite, characterised by a linear slope (SL). The contribution of each borehole to the total linear flow was stable with the depth of the borehole below the saprolite. This indicates the most productive part of the fractured layer of the aquifer. About 90% of the linear flow of the boreholes corresponded to this interval, for only 73% of the boreholes. The slope SL was 0.32 m/% or 32 m;

The third part corresponded to depths greater than 43 m. It shows a significant decrease in linear flow. This can be explained by the decrease in the number of producing fractures (see Section 3.3.2), and therefore by the decrease in the permeability at this depth of the aquifer, as shown in previous studies elsewhere in the world [20,24,37–39]. We are probably at the base of the fractured layer or at least in its least permeable section.

The graph for the lithological unit Biotite granodiorite and Biotite metagranodiorite (Figure 10b) shows:

The first part, at depths under about 27 m below the base of the saprolite, characterized by a rather steep slope (SL1) of 0.91 m/% or 91 m. About 27% of the linear discharge of the boreholes was found in this interval, for only 25% of the boreholes;

The second part of the curve had a linear slope in the interval [27–43 m[. The linear discharge was relatively stable within this interval. About 57% of the linear discharge of the boreholes was found in this interval, for only 24% of the boreholes. The slope SL2 was 0.28 m/%, or 28 m. Considered all together, the first and second part of the curve accounted for about 84% of the linear discharge of the boreholes, for only 62.2% of the boreholes;

The third part of the curve corresponded to depths greater than 43 m below the base of the saprolite. It shows a progressive decrease in the linear discharge, which is shown by a slope that increases with depth. We are therefore at the base of the fractured layer or at least at the least permeable part of it.

Of all the rock units studied in this work (see Figures S5, S9, S13 and S16 in Supplementary Materials), only the Biotite granodiorite and Biotite metagranodiorite unit has the first steeper SL1 slope in the depth interval delimited by "a and b". It is quite steep (0.91 m/%); however, it is less than the third part of the curve. The authors of [24] have also observed this slope but to a lesser extent (Figure 5), and without having described or interpreted it. This first section of the curve is described and interpreted in the next section of this paper (Section 3.3.4).

### 3.3.4. Characteristics of Shallow Biotite Granodiorite and Biotite Metagranodiorite Boreholes of Lower Linear Discharge

Interpretation of the distribution of linear discharges identified two distinct populations of boreholes within the single unit Biotite granodiorite and Biotite metagranodiorite (Figure 10b):

Population 1: consisting of boreholes with a depth below the base of the saprolite being less than or equal to "b" = 27 m;

Population 2: consisting of boreholes with a depth greater than "b" and lower than "c" = 43 m below the base of the saprolite.

Figure 11 shows, for each of the two borehole populations, five main parameters, consisting of well altitude (asl), well length, thickness of the saprolite, well length below the base of the saprolite, and instantaneous discharge.

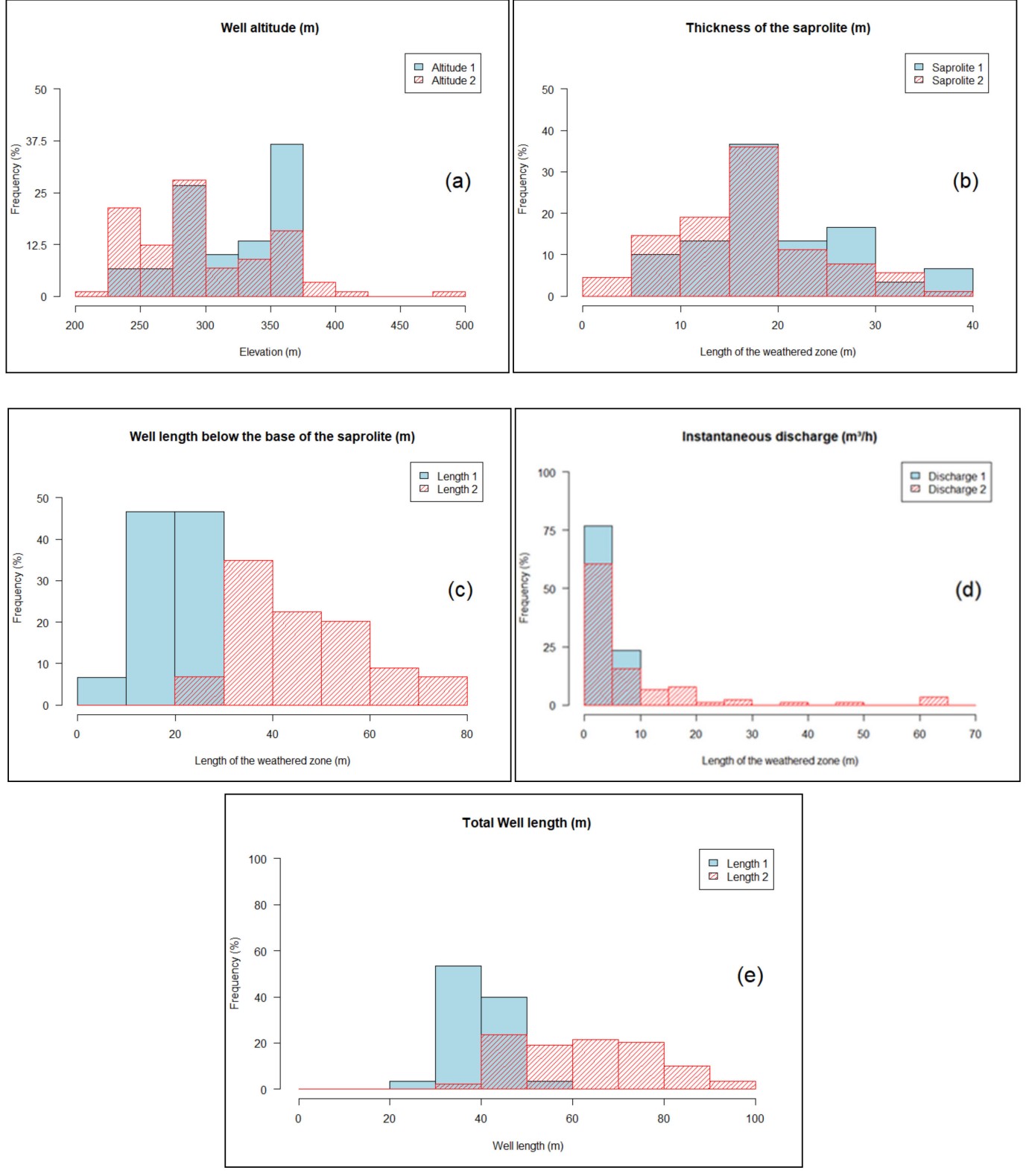

**Figure 11.** Histograms of altitude (**a**), thickness of the saprolite (**b**), well length below the base of the saprolite (**c**), instantaneous discharge (**d**), total well length (**e**) of the boreholes for the two populations considered (parameter 1 = population 1; N = 30 (depth ≤27 m), parameter 2 = population 2; N = 89, (27 < depth ≤43 m) for Biotite granodiorite and Biotite metagranodiorite (N = 119).

Table 7 shows the main statistical parameters of the lithological unit Biotite granodiorite and Biotite metagranodiorite.

**Table 7.** Statistical parameters in the Biotite granodiorite and Biotite metagranodiorite.

| Parameter | Population 1 (N = 30) | | | Population 2 (N = 89) | | |
|---|---|---|---|---|---|---|
| | Mean | Median | Standard Deviation | Mean | Median | Standard Deviation |
| Altitude (m) | 321.5 | 325 | 40 | 301 | 292 | 51.3 |
| Total well length (m) | 38.6 | 36 | 7.2 | 62.5 | 63 | 13.8 |
| Thickness of the saprolite (m) | 19.5 | 18 | 8 | 14.3 | 16.6 | 7.5 |
| Well length below the base of the saprolite (m) | 19 | 19.5 | 5.5 | 45.6 | 43.6 | 13 |
| Instantaneous discharge (m$^3$/h) | 3.5 | 3 | 2 | 8 | 3 | 12.7 |

Within this data set, it can be noted that there are no significant differences in the variation of the parameters studied between the two populations, with the exception of the parameter well length below the base of the saprolite of course, which is logical as population 1 is the closest from the base of the saprolite. The median values are similar for populations 1 and 2 (with the exception of the parameter well length below the base of the saprolite, which is logical by construction) (Table 7). However, the parameter well length is the only one which is very different for populations 1 and 2, with regard to the histograms, as well as the statistical parameters (mean, median, and also standard deviation, but for this latest parameter the difference is not significant, as the ratio standard deviation/mean remains similar, about 0.2 for both distributions).

Figure 12 shows the spatial distribution of the boreholes. There are no particularities that could differentiate the two populations. The boreholes of population 1 are located in both high and low altitude areas. The two populations are spatially confused. Similarly, there is no link between the populations and the two lithologies that have been grouped together.

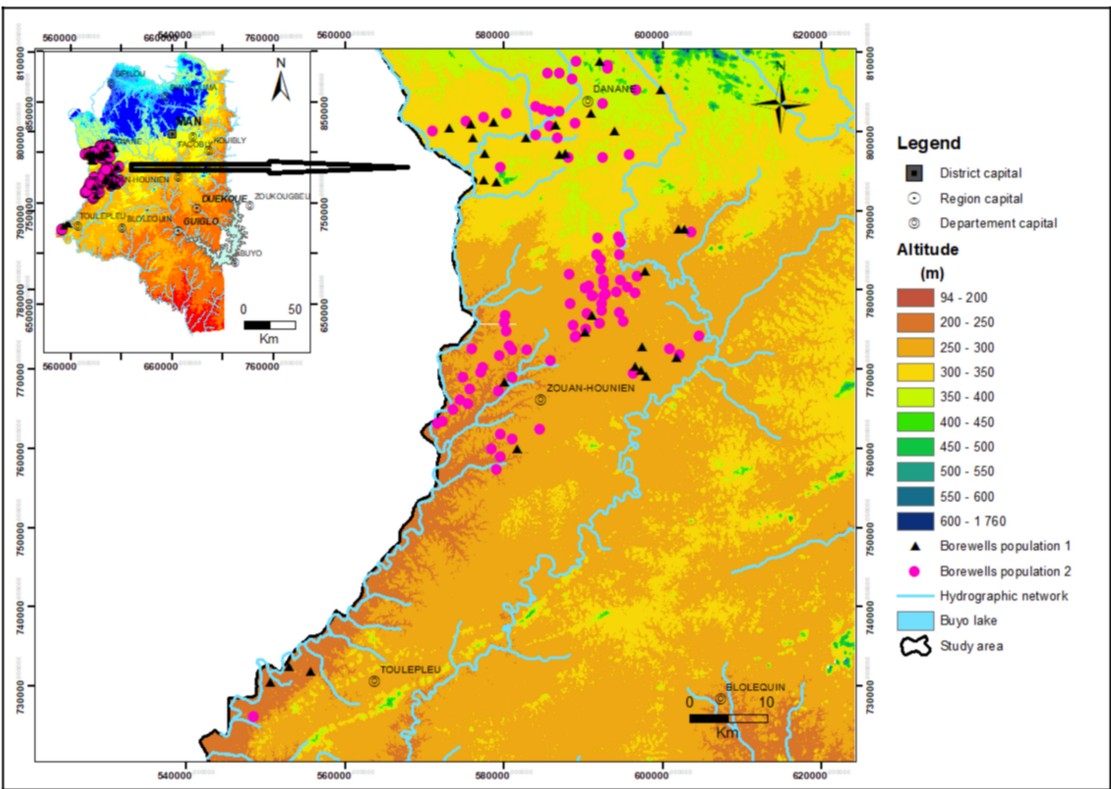

**Figure 12.** Spatial distribution of borewells population 1 (N = 30) and population 2 (N = 89) on the DEM for the Biotite Granodiorite and Biotite Metagranodiorite.

Thus, no robust explanation was found to explain the differentiation between populations 1 and 2 in this lithological unit Biotite granodiorite and Biotite metagranodiorite. The lower productivity could be due to a less developed or partly plugged fracture network just below the saprolite, but this is not consistent with the spatial distribution of the boreholes. As it is also a question of shallower drillings, there might be a bias linked to a specific drilling campaign, aiming at drilling low length boreholes, either with a low-power drilling machine or with a less powerful drilling machine if the aim was to drill shallow. In addition, such drillings might be drilled with a lower diameter, or even such machines might have had a lower air-lift capacity. Without additional data, these assumptions remain as hypotheses. As data about the dates of drilling and well diameters were lost during the civil war, it is not easy to check this hypothesis; a field survey (for both populations) may be devoted to measuring borehole diameters (at least tubes diameters) and/or to making video inspections.

In the rest of this research, populations 1 and 2 are grouped together to characterise the properties of the fractured layer of this lithological unit.

### 3.3.5. Properties of the Fractured Layer

In view of the above, it can be seen that the vertical distribution of linear discharge under the saprolite in all the lithological units highlights a breakpoint at which depth (40 to 45 m depending on the lithology) the linear discharge decreases significantly. This breakpoint describes the useful thickness of the fractured layer ($Lu$) beyond which the productivity of the aquifer decreases, statistically, very significantly. From a practical point of view, it means that if a bore well is dry at such depth the chance to get significant water strike below is, statistically, extremely low and that it is better to move the borehole to another place to tap water at a shallower depth. It can be seen that by applying the slope method, the useful thickness was not determined exclusively for 80% of cumulative percentage of linear discharge in the different lithological units of this study (Table 8).

Using another method proposed by [24] to evaluate this useful thickness, the 80 or 90% limit, is much less efficient for the same purpose; in fact, it appears arbitrary, as it does not consider the structure of the linear discharge distribution. Moreover, it introduces some dispersion in the results. Thus, it must be avoided.

The results obtained with the slope method for the different hard rock lithologies of the Montagnes District in Western Côte d'Ivoire (Table 8) show that the thickness of the most productive part of the fractured layer was between 40 and 45 m. The mean linear discharge and mean discharge showed a significant dispersion, between 0.25 and 0.34 m$^3$/h/m, and 8 and 15.4 m$^3$/h, respectively.

Figure 13 shows that there is no correlation between the slope and the corresponding mean discharge ($R^2 = 0.01$). This is logical, as the slope is highly dependent, notably, on the depth of the shallowest borehole below the base of the saprolite, which is not a hydrogeological parameter. Therefore, the slope should not be compared with the useful thickness ($Lu$) neither should it be compared with the mean linear discharge. The slope is thus not a quantitative indicator of the permeability of the fractured layer; it is only an indicator of its geometry. It cannot therefore be concluded that the higher the linear slope, the greater the discharge or vice versa. Consequently, it is not desirable to compare slopes to compare the permeability of different hard rock aquifers as argued in [24]. Instead, it should be accomplished by determining the useful thickness from the breakpoint of the cumulative linear discharges percentage curve, and by determining the linear average discharge from the population of boreholes considered and the corresponding average discharge. Nevertheless, and logically, it can be observed that the mean discharge was high (>5 m$^3$/h) in the first forty-five metres below the saprolite within all the different lithological units.

**Table 8.** Mean properties of the fractured layer evaluated with the slope method for the various hard rocks.

| Lithological Description | $a$ (m) | $b$ (m) | Length of the Most Permeable Part of the Fractured Zone Evaluated with the Slope Method (m/%) | Useful Thickness of the Fractured Layer $Lu$ (m) | Cumulative Percentage of Linear Discharge $Pq(Lu)$ (%) | Number of Wells $J$ ($Lu$) | Mean Linear Discharge of the Most Productive Part of the Fractured Layer $q_M(Lu)$ (m³/h/m) | Mean Discharge of the Most Productive Part of the Fractured Layer Part $Q_M(Lu)$ (m³/h) |
|---|---|---|---|---|---|---|---|---|
| Charnockitic orthogneiss and charnockites | 15 | 43 | 0.32 | 43 | 90 | 42 (29%) | 0.25 | 10.8 |
| Biotite granodiorite and Biotite metagranodiorite | 27 | 44 | 0.23 | 43 | 79 | 76 (63.8%) | 0.28 | 12.4 |
| Archean and Undifferentiated migmatites | 11 | 45 | 0.35 | 45 | 97 | 40 (77%) | 0.25 | 11.3 |
| Enderbites and Metaenderbites | 9 | 40 | 0.36 | 40 | 90 | 29 (61.7%) | 0.28 | 11.4 |
| Micashists, all Gneiss and Meta-arenites | 9 | 43 | 0.42 | 43 | 89 | 39 (83%) | 0.18 | 8 |
| Monzosyenites and Porphyroid granites | 8 | 45 | 0.45 | 45 | 89 | 25 (71.4%) | 0.34 | 15.4 |

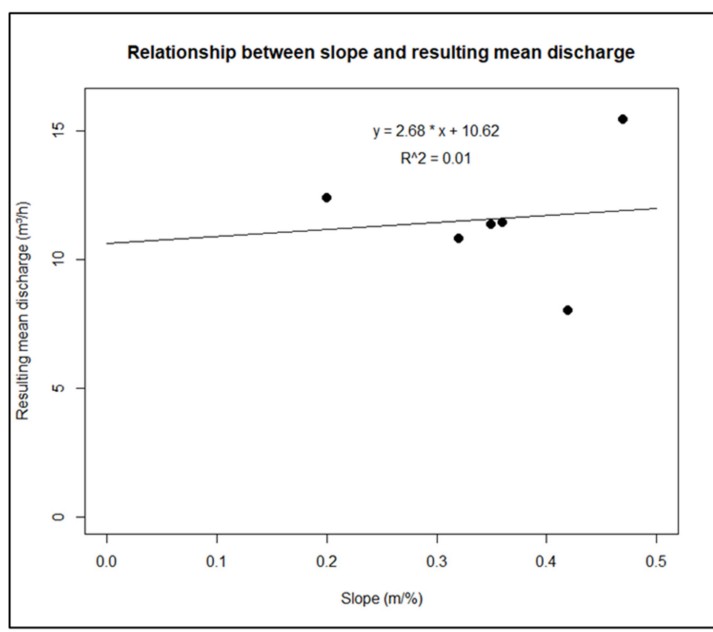

**Figure 13.** Relationship between the slope (X, m/%) and the resulting average discharge (Y, m$^3$/h) for all lithological units.

These results confirm that the fractured layer is thus an integral part of the weathering profile of the hard rock aquifers of the Montagnes District in Western Côte d'Ivoire, as in most regions of the world. In this area, it is characterised by a median useful thickness (*Lu*) of 43.5 m and a resulting median discharge of 11.3 m$^3$/h.

## 4. Discussion

The characterisation of hard rock aquifers has been the subject of several studies throughout the world, notably in Europe [1,6,20,26,43,44], Asia [27,38,39], America [11], and Africa [5,17,19,21,24,45–48]. Several methods (statistical methods, remote sensing-GIS, geostatistics, geophysics, lithological analysis, etc.) have been used in these studies to determine the geometric characteristics, structuring, and hydrodynamic properties of the layers constituting the weathering profile of hard rock aquifers. The permeability of an aquifer due to its fractured layer has been recognised by all authors, but its origin remains disputed by some hydrogeologists who attribute it to fractures resulting, among other things, from tectonic phenomena [1,26].

The present research was inspired by recent concepts of hard rock aquifers' permeability which argue that the fractured layer is permeable due to the presence of minerals such as biotite which heal during the weathering processes, then generate stresses in the rock that is at the origin of its fracturing, and thus at the origin of the fractured layer, hence causing the permeability of the hard rock aquifer [1,26]. Similar results were obtained by [16,49]. These authors showed that rock joints are not tectonic in origin but rather develop as a result of weathering processes, respectively, in hard rock and limestones. Statistical processing of ONEP borehole data based on the method of [24] has demonstrated the existence of such a fractured layer in the weathering profile of the hard rock aquifers of the Montagnes District in Western Côte d'Ivoire.

For all the lithological units studied, the location of the permeable stratiform fractured layer (SFL) under the saprolite was possible on the one hand by studying the variation in linear discharge as a function of drilling depth and on the other hand by studying the frequency of appearance of water strikes under the saprolite. This approach allowed us to observe that the linear discharge and the occurrence of water strikes decrease similarly with the depth under the saprolite. The highest linear discharges are obtained for shallow drilling depths under the saprolite (less than 40–45 m). More than 80% of the water strikes were observed in the first forty meters under the saprolite. This important variation is

followed by a decrease in the linear discharge and progressively the almost total absence of water strikes below this 40–45 m limit for all the lithologies encountered. This reduction means that we are at the base of the most productive part of the fractured layer. Data from water strikes suggest that this fractured layer is at maximum about 70 m thick.

Reference [17] observed in the Man-Danané region, from the distribution curve of water strikes as a function of depth, that there is a zone of productivity for water wells less than 100 m deep (total depth), located between 20 and 50 m below the base of the saprolite, beyond which water strikes become rare. These results from [17] are also consistent with previous work by [5,46,47]; however, these last three papers did not acknowledge, at that time, the role of weathering and attributed the fractures to tectonics.

The study of the distribution of water strikes in order to locate the fractured layer had not been used by [24,45,50] in their work to locate the fractured layer. The authors of [20] applied a similar approach but with reference to the ground surface, which is much less precise, while [45,50] used geophysical methods in addition to field observations to determine the geometric characteristics of this layer. Nevertheless, the results obtained in these studies are in line with ours.

The distribution of the permeability of the fractured layer and therefore of the hard rock aquifer, expressed as a cumulative percentage of linear discharge, is strongly correlated with the depth of the borehole under the saprolite. This distribution is characterised by two main sections on the cumulative curve highlighting two trends of linear discharge variation as a function of depth under the saprolite, as follows:

The first section between 0 and 40–45 m below the saprolite is characterised by high linear discharges on average. This means that we are in the fractured layer whose permeability is much higher than that of the overlying saprolite that has no productive fracture. These results are in agreement with the works of many authors in hard rocks, notably [20,24,37,38].

For low values (0 to 6 m) of drilling depth under the base of the saprolite, we note an absence of drilling data on all the lithological units (which is logical, the fractured layer being the target of the drillings). We also note a rather strong slope (lower average linear discharges) only in the case of Biotite metagranodiorite and Biotite granodiorite, between 6 and 15 m, for which no hydrogeological explanation was found. The main hypothesis is thus not really hydrogeological but that of an artefact due to smaller diameter borewells linked to drilling shallower wells with a less powerful drilling rig.

The section underlying the top one, below 45 m, is characterised by a progressive decrease in permeability until this reaches zero. This corresponds to the lowest part of the fractured layer where the density of fractures strongly decreases with depth. These results are in agreement with the work of [24,26].

Thus, for the six lithological units studied, representing 92% of the total area of the study area, the fractured layer is clearly identified as an integral part of the weathering profile of the hard rock aquifers in the study area. Its useful thickness is between 40 and 45 m whatever the lithology. The authors of [37] in Britany (France) showed that the most productive thickness of the useful fracture layer varies between 19 and 65.7 m (average 39.2 m), with an average of 36.1 m. The authors of [24] estimated it to be between 26 and 34 m in Burkina Faso. These values are globally higher than those obtained by [45] in the hard rock aquifers of Burkina Faso. According to these authors, the mean thickness of the fractured layer is 13.8 ± 8 m (maximum estimated thickness is about 20 m), and it varies between 2.5 and 35 m using the lithologs over all lithologies. With ERT, its maximum estimated thickness is about 20 m (it varies from 4 to 20 m). These values are globally lower than those obtained by [24] from hydrogeological data which are between 25 and 37 m on the same geological formations in Burkina Faso. The authors of [50,51] in Dimbokro (Central Côte d'Ivoire) showed that it varies respectively from 4 to 7 m in granites to 5 to 20 m in metasediments and volcanic rocks. The authors of [18] estimated it to be between 30 and 35 m on average in the Man-Danané region in Côte d'Ivoire, with the thickness of the saprolite being between 10 and 20 m (Table 9). The authors of [52] in the Tchologo

region (Northern Côte d'Ivoire) also estimated it to be between 40 and 45 m (considering the slope method—see below) in granitoid aquifers. The authors of [20] demonstrated in Central India (Hyderabad, Andhra Pradesh region) in an experimental granite basin, that the fractured layer has a limited vertical thickness of 35 m on average. The authors of [25] also demonstrated in India in a granitic terrain that the most transmissive zone is mainly between 9 and 35 m. The thickness of the fractured medium producing the highest discharge is between 20 and 30 m. Consequently, these authors also estimated that the thickness of the fractured layer is at maximum at about 35 m.

Given the decrease in the number of boreholes and inflows between 50 and 80 m (Figure 9a), it has been shown over the whole of the Montagnes region of Western Côte d'Ivoire that the fractured layer of the metamorphic hard rock aquifers could perhaps be estimated at around 80 m. The authors of [1,37] estimated in almost the same range of variation that the fractured layer of the basement aquifers was underlying the weathering layer and that its permeability was due to weathering processes.

All of this research provides a similar order of magnitude, whether the rocks are plutonic or metamorphic (our study focuses on these last rock types). These last less-studied rock types thus show here similar hydrogeological properties as the former. However, it is not excluded that other types of metamorphic rocks, such as highly schistosed rocks (micaschistes, hornfels, metasediments, metabasaltes, etc.), would provide different results.

These results are of prime importance for village hydraulics water well siting (see below). They are also very important for assessing the impact of a mining activity. We can now affirm from our results that the impact will mostly be limited to the topographic watershed area of the mine (only downstream it for quality impacts) and the watercourses that flow from it. In the case of the tectonic fracturing model, impacts could have been envisaged over much longer distances, for example, along kilometric scale regional fracturing corridors, although piezometric data should have been consistent with such an hypothesis (Aoulou et al., to be submitted).

Regarding the methodological aspects of this research, several outputs can be underlined:

First of all, this study again demonstrates the interest of constituting databases such as ONEP by gathering data from several projects. Even if they only comprise simple parameters, rather easy to measure on the field, their processing is very fruitful;

One must also underline that in the present study, the database was reconstituted by an administration of the central government after the civil war that occurred in this region of the country, as all local archives were destroyed. Our research shows that this reconstitution did not alter the quality of the database, or at least did not alter the quality of the most important data in the database;

The obtained results have direct practical implications. In combination with the saprolite thickness, they allow an estimation of the optimal drilling depth, i.e., the depth beyond which it is not necessary to drill because little discharge gain does not justify the increased drilling cost. Therefore, the results are of interest for the planning (duration) and financial costing of drilling campaigns (cumulative amount of length and depth of wells to be drilled, including the number of "dry" holes), including preliminary geophysical surveys, which can also be accurately sized. They provide an estimated minimum depth for the water wells required to penetrate the saprolite and reach the fractured layer, and can therefore be used to assess the drilling constraints, i.e., the technical characteristics of the drilling platform and wellbore equipment;

From a methodological standpoint also, this research shows that geological data from published documents such as geological maps or scientific papers must be preferred to the geological information available in the database. The latter was surely not collected by skilled geologists but was collected by the drillers or by hydrogeologists. This is thus surely not the reconstitution process after the war that altered this part of the data set;

This work also shows that a rigorous pre-treatment of the data is mandatory for avoiding any miscalculation and thus over-interpretation. For instance, duplicates appeared to constitute about 40% of the database;

From a methodological standpoint, our study definitely shows that the reference level for any computation in hard rock aquifer must be the base of the saprolite, and not at all the topographic surface;

It also shows that, when applying the method from [24], one must definitely use the slope method rather than the percentile method to define the useful thickness of the fractured layer. The latter appears to be purely arbitrary and is thus not relevant. Over all of the lithologies in our work, the median useful thickness is 43.5 m and corresponds to an average resulting discharge of 11.3 $m^3$/h. This useful thickness is obtained at the 89th to 97th percentile for the different lithological units, contrary to the work of [24] who considers that the useful thickness must be determined for 80% of the linear discharges. These authors argue that the 90th percentile method tends to slightly overestimate the thickness of the fractured layer compared with the slope method. Our work shows that it is not the case and that the percentage is lithology-/data set-specific. The choice of 80% or 90% of the linear discharges to estimate the geometric parameters of the fractured layer does not make sense; it should therefore be estimated on a case-by-case basis according to the shape of the vertical distribution of the linear discharges in percentage under the base of the saprolite;

It also shows that the method from [24] can be used with relatively limited data sets, here with boreholes populations between 35 and 145. [24] used it with much larger populations (several hundreds at least).

The mean linear discharge for the analysed dataset is between 0.18 and 0.34 $m^3$/h/m (the median is 0.26 $m^3$/h/m). The deduced mean discharge for the most productive part of the fractured layer is between 8 and 15.4 $m^3$/h. The authors of [52] demonstrated that the mean linear discharge and mean discharge for this useful thickness were 0.13 $m^3$/h/m and 5.21 $m^3$/h, respectively, in the granitoid aquifers. These values are similar to those obtained by [37], who demonstrated that, for a set of 42 geological formations, the linear discharge is between 0.078 and 0.84 $m^3$/h/m (average 0.38 $m^3$/h/m). They estimated that the discharge of the useful fractured layer varies from 3.8 to 42.6 $m^3$/h (Table 9). These results are therefore relatively close to those obtained in this study.

Thus, the results of the statistical treatment carried out on the basis of the method of [24] on the thickness of the saprolite (varying between 10 and 20 m) and the useful thickness of the most productive part of the fractured layer allow us to design a conceptual model describing the structure of the weathering profile of the hard rock aquifers in the study area (Figure 14). It is an adaptation of the model proposed by [26] from plutonic to metamorphic rocks.

**Table 9.** Geometric and linear discharge (thickness or depth range) of basement aquifers. NA: no data.

| References | Type of Rocks Studied | Studied Area | Total Thickness of the Fractured Layer (m) | Useful Thickness of the Fractured Layer Lu (m) | Mean Linear Discharge of the Fractured Layer $q_M(Lu)$ (m³/h/m) | Mean Discharge for Most Productive Part of the Fractured Layer $Q_M(Lu)$ (m³/h) |
|---|---|---|---|---|---|---|
| **Aoulou et al.** (this paper) | Metamorphic rocks: Charnockitic orthogneiss and Charnockites, Biotite granodiorite and Biotite metagranodiorite, Archean and Undifferentiated migmatites, Monzosyenites and Porphyroid granites, Enderbites and Metaenderbites, Micaschists, all Gneiss and Meta-arenites | Montagnes District in Western Côte d'Ivoire (31,000 km²) | 80 | 40–45 | 0.18–0.34 | 8–15.4 |
| **Kouamé et al. (2021)** [52] | Granitoids | Tchologo region in the North of Côte d'Ivoire (17,728 km²) | NA | 40–45 | 0.13 | 5.21 |
| **Soro et al. (2017)** [45] | Migmatite / Gneiss / Green rocks / Granites | Central northern Burkina Faso (14 km²) | 20 | NA | NA | NA |
| **Koita (2010)** [51] | Granites / Metasediments / Metavolcanites | Central eastern Ivory Coast (14 km²) | 80–100 | 40–60 | 0.03–3.7 | 1.8–2.75 |
| **Mougin et al. (2008)** [37] | Plutonic and Metamorphic rocks | French Britany (30,000 km²) | 75 | 19–65.7 | 0.078–0.84 | 3.8–42.6 |
| **Dewandel et al. (2006)** [20] | Granite | South India (50 km²) | 30–35 | 15–20 | NA | NA |
| **Maréchal et al. (2004)** [25] | Granite | South India (50 km²) | 35 | 15–30 | NA | NA |
| **Lasm (2000)** [18] | Granulitic gneiss / Migmatites / Amphibolo-pyroxenites / Charnockites / Orthogneiss | Montagnes region in Western of Côte d'Ivoire (area not specified in the manuscript) | 30–35 | 30 | NA | NA |

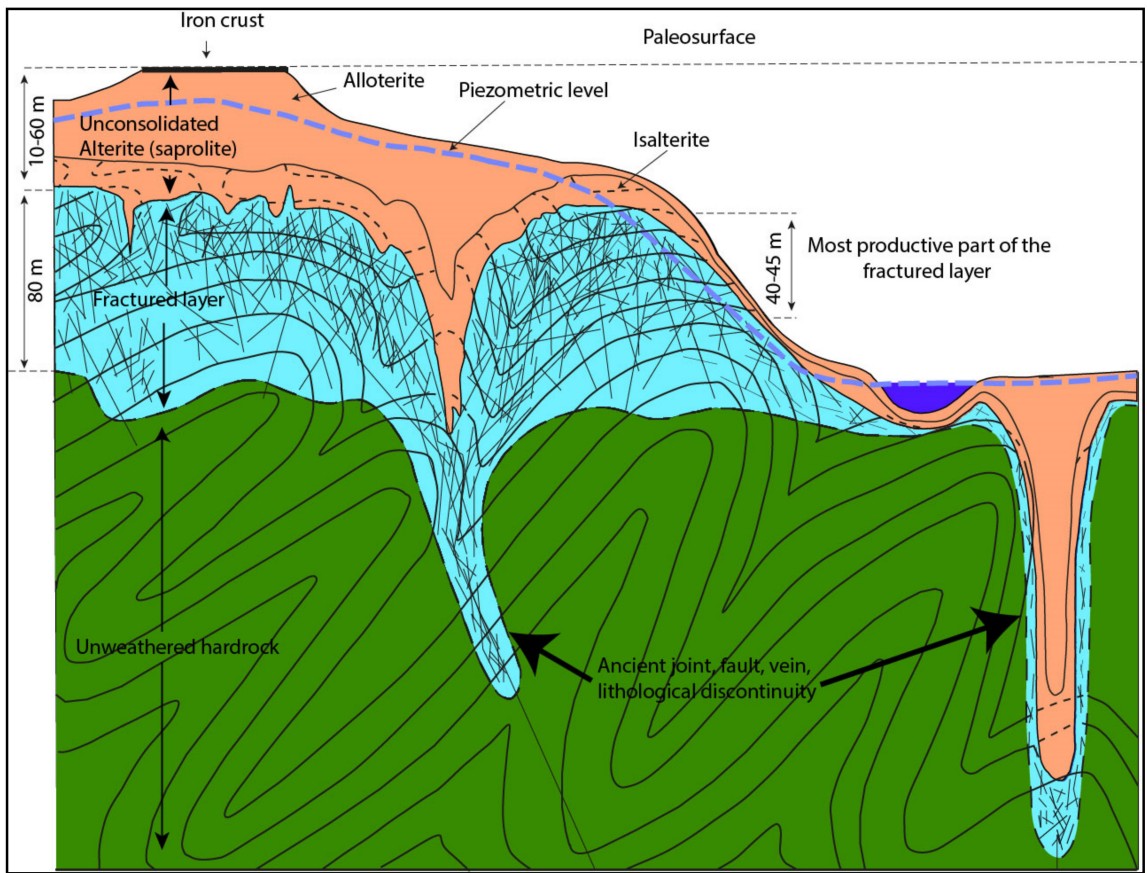

**Figure 14.** Conceptual model of the hard rock aquifers weathering profile in metamorphic rocks in the Montagnes District of Côte d'Ivoire (from [26], modified).

These results were obtained using the statistical method developed by [24] in Burkina Faso, as the geological formations studied had surely a similar geological history (the Eburnean orogenic cycle dated between 2400 and 1600 Ma [53,54]) and subsequent weathering and erosion phases. This proposed model is similar to the models presented by [26,43] in France, [50] in Côte d'Ivoire, and [20] in India. The authors of [17] underlined in the Man-Danané region the validity of such a model of a bilayer aquifer formed by alterites or alluvium overlying the fractured aquifer, although they did not provide precise estimates of the thickness of the various layers constituting it.

Other applications can be derived from the results, such as mapping the average discharge of boreholes on a regional/national scale for different depths and correlating these results with economic parameters such as drilling costs and such as the management of the groundwater resource or development plans of the groundwater resource. Other applications are beyond hydrogeology, such as urban and regional planning, soil engineering, quarrying prone areas, etc.

## 5. Conclusions and Perspectives

Statistical analysis of the ONEP borehole database, processing linear discharges and water strikes, allowed us to understand and characterize the structure, geometry, and hydrodynamic properties of the hard rock aquifers from the Montagnes District in Western Côte d'Ivoire. We observed that the structure of the aquifer is similar to the ones observed in several other areas in the world: the aquifer develops due to weathering processes, and it comprises the saprolite layer and an underlying fractured layer, respectively constituting the capacitive and transmissive parts of the aquifer, overlying the unweathered low-

permeable hard rock. This research shows that the fractured layer is here about 70 m thick but that its most productive zone is 40 to 45 m thick below the saprolite, depending on the lithology. The median thickness of the overlying saprolite mostly ranges between 10 and 20 m in all lithologies. The median productivity of the most productive part of the fractured layer is 11.3 m$^3$/h.

This work is a contribution to the characterization of metamorphic hard rock aquifers, which are far less known than plutonic ones. Metamorphic aquifers here show similar properties (thickness, hydrodynamic parameters) as plutonic ones.

This research also showed the interest of such borehole databases for research as well as practical purpose, notably for mapping the groundwater potential of an area and estimating the technical characteristics and costs of village hydraulic drilling campaigns and for assessing the impacts of anthropogenic activities, among which is mining. However, a rigorous pre-treatment of the data appears to be mandatory, and geological data from published maps or papers must be substituted for the data in the database. The research also allowed an improvement in the methodology used to process the boreholes' linear discharges, and notably showed that the slope method must definitely be preferred to the percentile method. This study allowed a good characterisation of the structure, the geometry of the weathering profile (thickness of the saprolite and thickness of the useful fractured layer), and the productivity of the aquifers (average linear discharges of the fractured layer). A next step is to characterise the hydrogeological functioning of these metamorphic aquifers, notably by studying their piezometry and hydrochemistry.

**Supplementary Materials:** The following are available online at https://www.mdpi.com/article/10.3390/w13223219/s1. Figure S1: Cumulative percentage of linear discharge (X axis, %; see definition of the linear discharge in the text) as function of the length of the well below the base of saprolite (Y axis, to estimate the properties of the fractured layer for the lithological (a) Biotite metagranodiorite (N = 105), (b) Biotite granodiorite (N = 14), and (c) Biotite metagranodiorite and Biotite granodiorite" (N = 119). Figure S2: Cumulative percentage of linear discharge (X axis, %; see definition of the linear discharge in the text) as function of the length of the well below the base of saprolite (Y axis, to estimate the properties of the fractured layer for the lithological: (a) Archean migmatite (N = 38), (b) Undifferentiated migmatite (N = 14), and (c) Archean and Undifferentiated migmatite (N = 52). Figure S3. Vertical distribution of all water strikes observed as a function of depth below the base of the saprolite in Archean and Undifferentiated migmatites (NWS1 + NWS2 = 64; with NWS1 = 44 and NWS2 = 20). Figure S4. Linearized discharges (X; m$^3$/h/m), computed with reference to well length below the base of the saprolite (Y: m), for Archean and undifferentiated migmatites (N = 52). Figure S5. Cumulative percentage of linear discharge to estimate the properties of the fractured layer for the lithology Archean and Undifferentiated migmatites (N = 52). Figure S6. Cumulative percentage of linear discharge (X axis, %; see definition of the linear discharge in the text) as function of the length of the well below the base of saprolite (Y axis, m) to estimate the properties of the fractured layer for the lithological (a) Metaenderbites (N = 41), (b) Enderbites (N = 6), and (c) Enderbites and Metaenderbites (N = 47). Figure S7. Vertical distribution of all water strikes observed as a function of depth below the base of the saprolite in Enderbites and Metaenderbites (NWS1 + NWS2 = 68; with N WS1 = 45 and N WS2, = 23). Figure S8. Linearized discharges (X; m$^3$/h/m), computed with reference to well length below the base of the saprolite (Y: m), for Enderbites and Metaenderbites (N = 47). Figure S9. Cumulative percentage of linear discharge to estimate the properties of the fractured layer for the lithology Enderbites and Metaenderbites (N = 47). Figure S10. Cumulative percentage of linear discharge (X axis, %; see definition of the linear discharge in the text) as function of the length of the well below the base of saprolite (Y axis, to estimate the properties of the fractured layer for the lithological: (a) Gneiss (N = 18), (b) Micaschists and Feldspathic gneiss (N = 11), and (c) Meta-arenites (N = 18) (d). Micaschists, all Gneiss and Meta-arenites (N = 47). Figure S11. Vertical distribution of all water strikes observed as a function of depth below the base of the saprolite in Micaschists, all Gneiss, and Meta-arenites (NWS1 + NWS2 = 59; with N WS1 = 41 and N WS2, = 18). Figure S12. Linearized discharges (X; m$^3$/h/m), computed with reference to well length below the base of the saprolite (Y: m), for Micaschists, all Gneiss, and Meta-arenites (N = 47). Figure S13. Cumulative percentage of linear discharge to estimate the properties of the fractured layer for the lithology Micaschists, all Gneiss, and Meta-arenites (N = 47). Figure S14. Vertical distribution of all

water strikes observed as a function of depth below the base of the saprolite in Monzosyenites and Porphyroid granites (NWS1 + NWS2 = 52; with N WS1 = 33 and N WS2, = 19). Figure S15. Linearized discharges (X; m$^3$/h/m), computed with reference to well length below the base of the saprolite (Y: m), for Monzosyenites and Porphyroid granites (N = 35). Figure S16. Cumulative percentage of linear discharge to estimate the properties of the fractured layer for the lithology Monzosyenites and Porphyroid granites (N = 35).

**Author Contributions:** K.A.A., S.P., P.L. and Y.M.S.O. designed the study, developed the methodology, and wrote the manuscript. K.A.A. collected the data and realized the processing of the data with S.P. and P.L., with the scientific support of B.D. All authors have read and agreed to the published version of the manuscript.

**Funding:** Our thanks go to the Scholarship Department of the Ministry of Higher Education and Scientific Research of Côte d'Ivoire for granting a scholarship to carry out this research work.

**Institutional Review Board Statement:** Not applicable.

**Informed Consent Statement:** Not applicable.

**Data Availability Statement:** Data available on request from the first author.

**Acknowledgments:** The authors are grateful to the National Office of Potable Water (ONEP), which is the guardian of the Ministry of Hydraulics in Côte d'Ivoire for providing the drilling database as part of this article. Our thanks also go to the Direction des bourses of the Ministry of Higher Education and Scientific Research of Côte d'Ivoire for granting a scholarship to carry out this research work. Madeleine Wheatley is also thanked for the translation of this paper from French to English. The three anonymous reviewers are also thanked for their careful review that helped to improve the manuscript.

**Conflicts of Interest:** The authors declare no conflict of interest.

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
