# Peer review of "Improving the Methods for Processing Hard Rock Aquifers Boreholes’ Databases. Application to the Hydrodynamic Characterization of Metamorphic Aquifers from Western Côte d’Ivoire"

_water, doi:10.3390/w13223219_

Round 1
Reviewer 1 Report
The paper addresses very good analysis for complex aquifer characterization using borehole data, which are often underutilized. The authors well explained their statistical approach, sorting out the borehole data and eliminating duplicate boreholes or the ones that lack required information. The results and outcomes of this study will be super useful to the local community and to the hydrogeology community as well. Most of the countries have similar borehole database and it should be utilized to address groundwater problems and parameters. In the meantime I would advise the authors to consider the following comments, if they see them useful:
- Reduce the length of the introduction and add a paragraph of literature review highlighting how others used borehole data base to do similar analysis. The reader needs to know where your research stands relative to others.
- I do not see a map showing the locations of the boreholes, especially the ones you used in your analysis after eliminating the low quality ones.
- I assume that the holes were drilled and described by different entities, if I am correct, how you dealt with the different style of borehole descriptions.
- I recommend the discussion section to be shortened.
Reviewer 2 Report
It would be interesting to continue the work, as the authors write at the end of Conclusions and Perspectives: "A next step will be to characterise the hydrogeological functioning of these metamorphic aquifers, notably by studying their piezometry and hydrochemistry".
Reviewer 3 Report
Good paper on databases in hydrogeology. Such tools are useful to both industry and researchers and I have produced some hydrogeological papers using the basic statistics from databases. However, all the comments need to be addressed before publication and there is my availability to review the manuscript a second time under request of the editor. Revise the language in Methods, Results and Discussion
Introduction
Lines 34-36. Large statement not backed up with references. I suggest the following papers with regards to fractured rock hydrogeology in an industrial and agricultural (including food production) framework. The papers proposed also follows your scheme going deeper in the subsurface: (i) capacitive saprolite, (ii) underlying transmissive fractured layer, overlying (iii) the unweathered impermeable layer
- Agbotui, P. Y., West, L. J., & Bottrell, S. H. (2020). Characterisation of fractured carbonate aquifers using ambient borehole dilution tests. Journal of Hydrology, 589, 125191.
- Medici, G., Baják, P., West, L. J., Chapman, P. J., & Banwart, S. A. (2021). DOC and nitrate fluxes from farmland; impact on a dolostone aquifer KCZ. Journal of Hydrology, 595, 125658.
- Worthington, S. R., Davies, G. J., & Alexander Jr, E. C. (2016). Enhancement of bedrock permeability by weathering. Earth-Science Reviews, 160, 188-202.
Lines 33-109. I suggest adding other databases on borehole information. An example is Allen et al. 1997 in England and Wales that shows basic statistics on aquifer properties. Your database might be the first one that introduce the concept of “effective thickness”. I would highlight this original aspect
- Allen, D. J., Brewerton, L. J., Coleby, L. M., Gibbs, B. R., Lewis, M. A., MacDonald, A. M., ... & Williams, A. T. (1997). The physical properties of major aquifers in England and Wales.
Line 100. Be more explicit on aim and objectives of your study. Use numbers for the multiple objectives
Materials and methods
Lines 102-103. Revise language
Lines 276-279. Fix problems of formatting
Line 288. Avoid using a reference in a title. Move Courtois et al. (2010) below
Results
Lines 420, 433, 489-491, 609. Do not use bold and re-ward the sentences in a form suitable for a scientific paper
Line 465. Avoid to start a sentence with “it”. Please, revise the language
Discussion
Line 711. Add literature on origin of rock joints of tectonic and non-tectonic origin that affect groundwater flow
- Medici, G., West, L. J., Mountney, N. P., & Welch, M. (2019). Permeability of rock discontinuities and faults in the Triassic Sherwood Sandstone Group (UK): insights for management of fluvio-aeolian aquifers worldwide. Hydrogeology Journal, 27(8), 2835-2855.
- Hencher, S., & Knipe, R. (2007, July). Development of rock joints with time and consequences for engineering. In 11th ISRM Congress. OnePetro.
Conclusions
“We obtained….to weathering”. Revise the language and avoid using “that” so many times
Figures
The figures are to many and in some cases can be combined.
- Figures S1 and S2 can be combines using “a” and “b”. This is only an example that can be repeated
References
Integrate the bibliography with the list of relevant papers above
My best wishes,
